# Conventional sub-soil irrigation techniques do not lower carbon emissions from drained peat meadows

Stefan Theodorus Johannes Weideveld[1*], Weier Liu[2], Merit van den Berg[1], Leon Peter Maria Lamers[1], Christian Fritz[1],

[1] - Aquatic Ecology and Environmental Biology, Institute for Water and Wetland Research, Radboud University, Heyendaalseweg 135, 6525, AJ, Nijmegen, the Netherlands.

[2] - Integrated Research on Energy, Environment and Society, University of Groningen, Nijenborgh 6, 9747 AG, Groningen, the Netherlands

*Corresponding author

E-mail addresses: *Stefan.Weideveld1@gmail.com ,S.Weideveld@science.ru.nl (S.T.J. Weideveld)*

## Abstract

The focus of current water management in drained peatlands is to facilitate optimal drainage, which has led to soil subsidence and a strong increase in greenhouse gas (GHG) emissions. The Dutch land and water authorities proposed the application of sub-soil irrigation (SSI) system on a large scale to potentially reduce GHG emissions, while maintaining high biomass production. Based on model results, the expectation was that SSI would reduce peat decomposition in summer by preventing groundwater tables (GWT) to drop below -60 cm. In 2017–2018, we evaluated the effects of SSI on GHG emissions ($CO_2$, $CH_4$, $N_2O$) for four dairy farms on drained peat meadows in the Netherlands. Each farm had a treatment site with SSI installation and a control site drained only by ditches (ditch water level -60/-90 cm, 100 m distance between ditches). The SSI system consisted of perforated pipes -70 cm from surface level with spacing of 5–6 m to improve drainage during winter-spring and irrigation in summer. GHG emissions were measured using closed chambers every 2–4 weeks for $CO_2$, $CH_4$ and $N_2O$. Measured ecosystem respiration ($R_{eco}$) only showed a small difference between SSI and control sites when the GWT of SSI sites were substantially higher than the control site (> 20 cm difference). Over all years and locations, however, there was no significant difference found, despite the 6–18 cm higher GWT in summer and 1–20 cm lower GWT in wet conditions at SSI sites. Differences in mean annual GWT remained low (< 5 cm). Direct comparison of measured $N_2O$ and $CH_4$ fluxes

between SSI and control sites did not show any significant differences. $CO_2$ fluxes varied according to temperature and management events while differences between control and SSI sites remained small. Therefore, there was no difference between the annual gap-filled net ecosystem exchange (NEE) of the SSI and control sites. The net ecosystem carbon balance (NECB) was on average 40 and 30 t $CO_2$ ha$^{-1}$ yr$^{-1}$ in 2017 and 2018 on the SSI sites and 38 and 34 t $CO_2$ ha$^{-1}$ yr$^{-1}$ in 2017 and

2018 on the control sites. This lack of SSI effect is probably because the GWT increase remains limited to deeper soil layers (60–120 cm depth), which contribute little to peat oxidation.

We conclude that SSI modulates water table dynamics but fails to lower annual carbon emission. SSI seems unsuitable as a climate mitigation strategy. Future research should focus on potential effects of GWT manipulation in the uppermost organic layers (-30 cm and higher) on GHG emissions from drained peatlands.

**1 Introduction**

Peatlands cover only 3% of the land and freshwater surface of the planet, yet they contain one third of the total carbon (C) stored in soils (Joosten and Clarke, 2002). Natural peatlands capture C by producing more organic material than decomposed due to waterlogged conditions (Gorham et al., 2012;Lamers et al., 2015). Drainage of peatlands for agricultural purposes leads to aerobic oxidation of organic material and increased gas exchange releasing $CO_2$ and $N_2O$ at high rates (Regina et al.,

2004;Joosten, 2009;Hoogland et al., 2012;Lamers et al., 2015;Leifeld and Menichetti, 2018). Soil subsidence occurs when the groundwater table (GWT) drops through drainage, leading to physical and chemical changes of the peat including microbial breakdown of organic matter. This results in consolidation, shrinkage, compaction and increased decomposition (Stephens et al., 1984;Hooijer et al., 2010). Soil subsidence increases the risk of flooding (frequency and duration) in areas where soil surface subsides below river and sea levels (Syvitski et al., 2009). In the Netherlands, 26% of the surface area is currently

below sea level, an area currently inhabited by 4 million people (Kabat et al., 2009). This area is expected to increase due to further land subsidence, while sea level is rising at the same time, which is a general issue of coastal peatlands (Erkens et al., 2016;Herrera-García et al., 2021). Additionally, peatland subsidence alters hydrology on various scales, which lead to frequent drainage failure, saltwater intrusion and loss of productive lands (Dawson et al., 2010;Herbert et al., 2015). Ongoing peatland

subsidence inflict high societal costs and difficulties in maintaining productive land use (Van den Born et al., 2016;Tiggeloven

et al., 2020).

The peatland area used for agriculture is estimated at 10% for the USA and 15% for Canada, and varies from less than 5 to more than 80% in European countries (Lamers et al., 2015). In the Netherlands, 85% of the peatland areas are in agricultural use (Tanneberger et al., 2017), leading to $CO_2$ emissions of 7 Mt $CO_2$-eq per year, accounting for >25% of total greenhouse

gas (GHG) emissions from Dutch agriculture (Arets et al., 2020). Fundamental changes in the management of peatlands are required if land use, biodiversity and socio-economic values including GHG emission reduction are to be maintained.

$CO_2$ emissions from peatlands are related to the GWT position below surface, which affects oxygen intrusion, moisture content and temperature. There is ample evidence that elevating GWT to 0-20 cm below surface results in substantial reduction of $CO_2$

emissions from (formerly) managed peatlands (Hendriks et al., 2007;Hiraishi et al., 2014;Jurasinski et al., 2016;Tiemeyer et al., 2020). Increasing GWT close to the surface does not only constrain aerobic $CO_2$ production and rapid gas exchange but also reduces land-use intensity (fertilization, tillage, planting, grazing). Additionally, high GWT could favor vegetation assemblages with a higher carbon sequestration potential (e.g. peat forming plants) compared to common fodder grasses and crops. Experimental studies on water table manipulation stressed the importance of rewetting the upper 20-30 cm to achieve

noteworthy $CO_2$ emissions reduction (Regina, 2014;Karki et al., 2016) which seems in line with the correlation of $CO_2$ emissions with GWT based on a meta-analysis of field $CO_2$ emission data by Tiemeyer et al. (2020).

Dutch water- and land-authorities have relied on ground surface elevation measurements to estimate peat loss rather than $CO_2$ flux measurements to calculate $CO_2$ emissions from peatlands (Arets et al., 2020) and the effects of elevated GWT on $CO_2$

emissions. Two assumptions are generally made when inferring surface elevation data into $CO_2$ emission from surface elevation changes: 1) Elevation changes are directly related to C losses from peatlands within a time frame of years ignoring physical changes of peat following drainage. As a conversion factor 2.23 t $CO_2$ ha$^{-1}$ per mm subsidence is assumed (Kuikman et al., 2005;Van den Akker et al., 2010). 2) The average lowest summer GWT (GLG) is assumed to be a major control factor

of subsidence rates of peat surface elevation and henceforth $CO_2$ emissions based on the first assumption above (Arets et al., 2020). As a consequence of both assumptions, Dutch climate mitigation frameworks focus on elevating summer GWT in peatlands rather than mean annual GWT (Querner et al., 2012;Brouns et al., 2015). Dutch water- and land-authorities expect that increasing the average lowest summer GWT by 20 cm would result in an emission reduction equalling 10.5 t $CO_2$ ha$^{-1}$ yr$^{-1}$ (Van den Akker et al., 2007;Brouns et al., 2015;Van den Born et al., 2016).

The use of subsoil irrigation and drainage systems (SSI) have been proposed to elevate summertime GWT and thereby presumably reducing $CO_2$ emissions (Van den Akker et al., 2010;Querner et al., 2012). SSI works by installing drainage/irrigation pipes at around 70 cm below the surface or at least 10 cm below the ditch water level. Water from the ditch can infiltrate into the peat adjacent to SSI pipes and thereby limit GWT drawdowns during summer (c.f. (Hoving et al., 2013). Next to irrigation, SSI pipes primarily fulfill a drainage function when the GWT is above the ditch water level. Based on the elevating effects on summer groundwater table SSI was assumed to reduce of C emissions from peatlands by 50% according to the soil-carbon-water model (Querner et al., 2012;Van den Born et al., 2016). However, th effect of SSI on C emissions has nog yet been tested by field measurements of C-fluxes.

The aim of our study was to quantify the effects of SSI on the GWT and GHG emissions, with consideration of the farm field net ecosystem carbon balance (NECB). We questioned 1) to what extent can SSI regulate GWT, especially during dry conditions in summer, 2) whether the SSI can substantially reduce (up to 50% as assumed by authorities) $CO_2$ emission compared to traditional ditch drainage. To adress these questions we directly compared GHG emissions from a control grassland (traditional ditch drainage) with a treatment grassland (SSI) on four farms over a periode of 2 years.

## 2 Material and methods

### 2.1 Study area

The study areas are located in a peat meadow area in the province of Friesland, the Netherlands. The climate is humid Atlantic with an average annual precipitation of 840 mm and an average annual temperature of 10.1°C (The Royal Netherlands Meteorological Institute, KNMI, reference period 1999-2018). About 62% of the Frisian peatland region is now used as grassland for dairy farming (Hartman et al., 2012). Agricultural land in Friesland is farmed intensively, with high yields, and intensive fertilization (>230 kg N ha$^{-1}$ yr$^{-1}$), combined with wide fields with deep drainage. One third of the fields are drained to -90 – -120 cm below soil surface. Large parts of these grasslands are covered with a carbon rich clay layer, ranging from 20–40 cm thick. The peat layer below has a thickness of 80–200 cm, which consists of sphagnum peat on top of sedge, reed and alder peat. The top 30 cm of the peat layer is strongly humified (van Post H8-H10) and the peat below 60 – 70 cm deep is only moderately decomposed (van Post H5-H7). On two locations (C and D, see below), there is a 'schalter' peat layer present, which is highly laminated peat (compacted/ hydrophobic layers of *Sphagnum cuspidatum* remnants) with poor degradability and poor water permeability. The grasslands were dominated by *Lolium perenne*; other species such as *Holcus lanatus*, *Elytrigia repens*, *Ranunculus acris* and *Trifolium repens* were present in a low abundance in 2017-2019.

**Table 1 Soil and land-use characteristics of the research sites in the Frisian peat meadows, the Netherlands. Averages per soil type, gravimetric soil moisture content taken August 2017, Dry bulk density, Organic matter content, and elemental Carbon content.**

| Farm | Treatment | Field size | peat thickness | Soil type | Soil depth | Soil moisture | Bulk density | Organic matter | Carbon content | Carbon content |
|------|-----------|-----------|----------------|-----------|------------|---------------|--------------|----------------|----------------|----------------|
| | | ha | m | | cm | % | g DW/cm3 | g Org/L | g C/l | g C/kg |
| A ) Organic Grazing | SSI | 2 ha | 1.6 m | Mineral | 0 − 35 | 38.1 | 0.99 | 123 | 52 | 53 |
| | | | | Peat | 35 − 60 | 77.1 | 0.23 | 144 | 77 | 335 |
| | | | | Peat | 60 − 80 | 82.1 | 0.14 | 130 | 68 | 485 |
| | Control | 0.6 ha | 2 m | Mineral | 0 − 40 | 37.6 | 0.93 | 130 | 54 | 58 |
| | | | | Peat | 40 − 60 | 59.2 | 0.24 | 156 | 83 | 345 |
| | | | | Peat | 60 − 80 | 85.3 | 0.16 | 154 | 98 | 613 |
| B ) | SSI | 2.3 ha | 1.4 m | Peat | 0 − 20 | 51 | 0.44 | 270 | 108 | 245 |

| | | | | | | | | | | |
|---|---|---|---|---|---|---|---|---|---|---|
| Conventional | | | | Peat | 20 − 60 | 79.3 | 0.19 | 169 | 77 | 403 |
| Grazing | | | | Peat | 60 − 80 | 88.4 | 0.12 | 118 | 60 | 499 |
| | Control | 2.3 ha | 1.4 m | Peat | 0 − 20 | 50.1 | 0.49 | 273 | 138 | 282 |
| | | | | Peat | 20 − 60 | 77.7 | 0.17 | 141 | 72 | 424 |
| | | | | Peat | 60 − 80 | 86.5 | 0.13 | 122 | 67 | 515 |
| C | SSI | 1.2 ha | 1.3 m | Mineral | 0 − 30 | 36 | 0.71 | 128 | 58 | 82 |
| Conventional | | | | Schalter | 30 − 40 | 79.2 | 0.19 | 177 | 88 | 461 |
| Mowing | | | | Peat | 40 − 60 | 82.2 | 0.18 | 129 | 64 | 357 |
| | | | | Peat | 60 − 80 | 87.5 | 0.11 | 133 | 81 | 740 |
| | Control | 1.8 ha | 1 m | Mineral | 0 − 30 | 38 | 0.75 | 142 | 59 | 79 |
| | | | | Schalter | 30 − 40 | 78.7 | 0.19 | 177 | 92 | 486 |
| | | | | Peat | 40 − 60 | 84.3 | 0.12 | 116 | 60 | 499 |
| | | | | Peat | 60 − 80 | 89.2 | 0.1 | 134 | 72 | 715 |
| D | SSI | 2.4 ha | 0.9 m | Mineral | 0 − 30 | 37.7 | 0.85 | 155 | 74 | 87 |
| Conventional | | | | Schalter | 30 − 40 | 63.9 | 0.3 | 267 | 85 | 284 |
| Mowing | | | | Peat | 40 − 60 | 84.3 | 0.19 | 137 | 73 | 385 |
| | | | | Peat | 60 − 80 | 80.2 | 0.14 | 130 | 55 | 390 |
| | Control | 3.5 ha | 0.9 m | Mineral | 0 − 25 | 32.9 | 0.82 | 141 | 73 | 89 |
| | | | | Schalter | 25 − 35 | 70 | 0.27 | 173 | 86 | 318 |
| | | | | Peat | 35 − 60 | 84.1 | 0.15 | 142 | 83 | 551 |
| | | | | Peat | 60 − 80 | 81.9 | 0.11 | 109 | 70 | 632 |


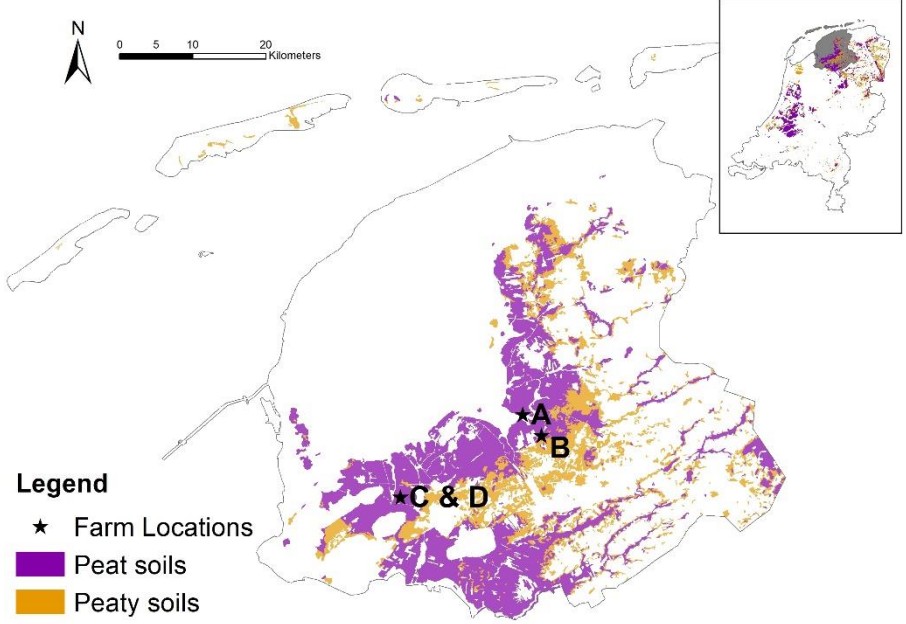

**Figure 1 Soil map with field locations situated in the province of Friesland, the Netherlands. Peat soils refer to soils with an organic layer of at least 40 cm within the first 120 cm, while peaty soils are soils with an organic layer of 5-40 cm within the first 80**
**cm.**

## 2.2 Experiment setup

Four sites were set up at dairy farms with land management and soil types representative for Friesland (see Table 1 and Fig.

1). Each location consisted of a treatment site with SSI pipes and a control site. The SSI pipes were installed at a depth of 70

cm below the surface and 5–6 m apart from each other, except for the D location where pipes were 5 m apart. The overall

drainage intensity was around 2000 m ha$^{-1}$. The pipes were either directly connected to the ditch (A and C) or connected to a

collector tube that was connected to a ditch (B and D). The connections with ditches were placed 10 cm below the targeted

ditchwater level that was regulated by a complex network of water inlet and pumps at the lowest parts of the polder. The

control sites are fields that have traditional drainage with deep drainage ditches with convex fields and small shallow ditches

(furrows).

On the treatment sites, three gas measurement frames 80 × 80 cm were placed for the duration of the experiment on 0.5 m, 1.5

m and 3 m distance from the chosen SSI pipe (Fig. 2), representing best the variation in the environmental conditions and

vegetation. The control sites were located 32 – 42 m from the ditch. Dip well tubes were installed to monitor water tables 0.5,

1.5 and 3 m from the pipe, pairing with the locations of gas measurement frames (Fig. 2). The nylon coated tubes were 5 cm wide and perforated filters (130-150 cm length) were placed in the peat layer. The tube 1.5 m from the SSI pipe was equipped with a pressure sensor and a data logger (ElliTrack-D, Leiderdorp instruments, Leiderdorp, Netherlands) that measures and records the GWT every hour. Ten more dip well tubes were further placed at intervals 0.5 and 3 m from the pipes in the field, which were manually sampled every 2 weeks during gas sampling campaigns, to obtain the variation on field scale.

To determine soil properties, soil samples were taken using a gouge auger in three replicates till 0.8 m depth, 1.5 meter from the SSI pipes taken in august 2017. For soil moisture, sediment samples were weighed and subsequently oven-dried at 105°C for 24 h. Organic matter content was determined via loss on ignition. Dried sediment samples were incinerated for 4 h at 550°C (Heiri et al., 2001). Total nitrogen (TN) and total carbon (TC) was determined in soil material (9–23 mg) using an elemental CNS analyzer (NA 1500, Carlo Erba; Thermo Fisher Scientific, Franklin, USA).


    Soil temperature at -5, -10 and -20 cm depth were continuously measured (12-Bit Temperature sensor -S-TMB-M002, Onset Computer Corporation, Bourne, USA) during the run time of the experiment and recorded every 5 minutes on a data logger (HOBO H21-USB Micro Station Onset Computer Corporation, Bourne, USA). Because of the frequent failure of sensors, extra temperature sensors (HOBO™ pendant loggers, model UA-002-64, Onset Computer Corporation, Bourne, USA) were

placed in the soil at a depth of -5 and -10 cm.

    At farms A and D, sensors were set up at 1.5 m above ground to measure photosynthetically active radiation (PAR, Smart Sensor S-LIA-M003, ONSET Computer Corporation, Bourne, USA), air temperature and air relative humidity (Temperature/Relative Humidity Smart Sensor, S-THB-M002, Onset Computer Corporation, Bourne, USA). Data were logged

every 5 minutes (HOBO H21-USB Micro Station, Onset Computer Corporation, Bourne, USA). Average air temperature and precipitation from the weather station Leeuwarden (18 to 30 km distance from research sites) were used (KNMI). The location specific precipitation was estimated using radar images with a resolution of $3 \times 3$ km.

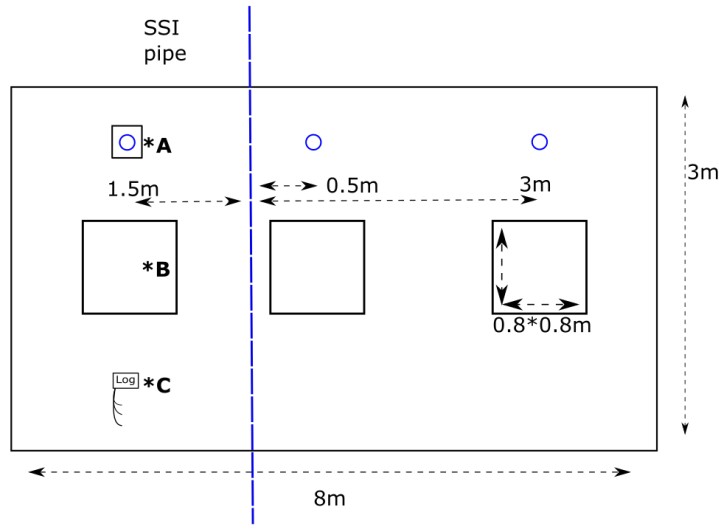

**Figure 2 Overview field site SSI. Blue dashed line = SSI pipe, blue circle = dipwell, \*A – dipwell with data logger, \*B – greenhouse gas flux measurement frame, \*C – data logger, -5 -10 -20 soil temperature**

C-export from frames used GHG measurements was determined by harvesting the standing biomass eight times in 2017 and five times in 2018. Two of the harvest moments in 2017 were extra planned, once in May because of the fast grass growth and grass height exceeding 30 cm, and the other in December in order to reset the grass height to the start of the experiment for next year. Surrounding the frames an area of 8 x 3 m was fenced off to avoid disturbance from grazing and other field activities (Fig. 2). The fenced-off area outside the frames was managed with 5 cuts per year to have a similar grass height with the

farmland. The biomass was harvested, weighed for fresh weight and dried at 70 °C until constant weight. Total nitrogen (TN) and total carbon (TC) was determined in dry plant material (3 mg) using an elemental CNS analyzer (NA 1500, Carlo Erba; Thermo Fisher Scientific, Franklin, USA). Due to grazing disturbance in 2018, an estimation instead of measurements was made for the C-export of location A in consultation with the farmer, but excluded from statistical analysis. Four times per year slurry manure from location C was applied to all plots. The slurry was diluted with ditchwater (2:1 ratio) and applied above

ground in the gas measurement frames and the surrounding area (119 – 181 kg N ha$^{-1}$ yr$^{-1}$ for 2017 and 129 – 162 kg N ha$^{-1}$ yr$^{-1}$ for 2018 with a C/N ratio of 16.3±1.3).

## 2.3 Flux measurements

$CO_2$ exchange was measured from January 2017 to December 2018, at a frequency of two measurement campaigns a month during growing season (April – October) and once a month during winter. This resulted in 34 (A), 35 (C and D) and 38 (B) campaigns over the two years for $CO_2$ and $CH_4$. The $N_2O$ emissions were measured with a lower frequency with 22 (A), 20 (B and C) and 17 (D) campaigns over the two years. A measurement campaign consisted of flux measurements with opaque (dark) and transparent (light) closed chambers (0.8x0.8x0.5 m) to be able to distinguish ecosystem respiration ($R_{eco}$) and gross primary production (GPP) from net ecosystem exchange (NEE). An average of 9 light and 10 dark measurements during winter, and 18 light and 20 dark measurements during summer, were carried out over the course of the day, to achieve data over a gradient in soil temperature and PAR.

The chamber was placed on a frame installed into the soil and connected to a fast greenhouse gas analyzer (GGA) with cavity ring-down spectroscopy (GGA-30EP, Los Gatos Research, Santa Clara, CA, USA) to measure $CO_2$ and $CH_4$ or to a G2508 gas concentration analyzer with cavity ring-down spectroscopy (G2508 CRDS Analyzer, Picarro, Santa Clara, CA, USA) to measure $N_2O$. To prevent heating and to ensure thorough mixing of the air inside the chamber, the chambers were equipped with two fans running continuously during the measurements. For $CO_2$ and $CH_4$, each flux measurement lasted on average 180s. $N_2O$ fluxes were measured on all frames at least once during a measurement campaign, with an opaque chamber for 480s per flux.

PAR was manually measured (Skye SKP 215 PAR Quantum Sensor, Skye instruments Ltd, Llandrindod Wells, United Kingdom) during the transparent measurements, on top of the chamber. The PAR value was corrected for transparency of the chamber. Within each measurement, a variation in PAR higher than 75 $\mu$mol m$^{-2}$ s$^{-1}$ would lead to a restart of the measurement. Soil temperature was measured manually in the frame after the dark measurements at -5 and -10 cm depth (Greisinger GTH 175/PT Thermometer, GMH Messtechnik GmbH, Regenstauf, Germany). Grass height was measured using a straight scale with a plastic disk with a diameter of 30 cm before starting the measurement campaign.

**2.4 Data analyses**

**2.4.1 Flux calculations**

Gas fluxes were calculated using the slope of gas concentration over time (Almeida et al., 2016) (eq.1).

$$F = \frac{V}{A} \times slope \times \frac{P \times F1 \times F2}{R \times T}$$

                                                                                                                                                                                              (1)

Where F is gas flux (mg m$^2$ d$^{-1}$), V is chamber volume (0.32 m$^3$), A is the chamber surface area (0.64 m$^2$), slope is the gas

concentration change over time (ppm second$^{-1}$); P is atmospheric pressure (kPa); F1 is the molecular weight, 44 g mol$^{-1}$ for

$CO_2$ and $N_2O$ and 16 g mol$^{-1}$ for $CH_4$; F2 is the conversion factor of seconds to days; R is gas constant (8.3144 J K$^{-1}$ mol$^{-1}$);

and T is temperature in Kelvin (K) in the chamber.

**2.4.2 R$_{eco}$ modeling**

To gap-fill for the days that were not measured for an annual balance for $CO_2$ exchange, R$_{eco}$ and GPP models needed to be

fitted with the measured data for each measurement campaign. R$_{eco}$ was fitted with the Lloyd-Taylor function (Lloyd and

Taylor, 1994) based on soil temperature (Eq. 2):

$$R_{eco} = R_{eco,Tref} \times e^{E_0\left(\frac{1}{T_{ref}-T_0} - \frac{1}{T-T_0}\right)}$$

                                                                                                                                                                                              (2)

Where R$_{eco}$ is ecosystems respiration, R$_{eco,Tref}$ is ecosystem respiration at the reference temperature (T$_{ref}$) of 281.15 K and was

fitted for each measurement campaign, E$_0$ is long term ecosystem sensitivity coefficient (308.56, (Lloyd and Taylor, 1994)),

T$_0$ Temperature between 0 and T (227.13, Lloyd and Taylor, 1994), T is the observed soil temperature (K) at 5 cm depth and

T$_{ref}$ is the reference temperature (283.15 K). If it was not possible to get a significant relationship between the T and the R$_{eco}$

with data from a single campaign, data were pooled for two measuring days to achieve significant fitting (Beetz et al.,

2013;Poyda et al., 2016;Karki et al., 2019)

**2.4.3 GPP modeling**

GPP was obtained by subtracting the measured R$_{eco}$ ($CO_2$ flux measured with the dark chambers) from the measured NEE

($CO_2$ flux measured with the light chambers). For the days in between the measurement campaigns, data were modeled with

the relationship between the GPP and PAR using a Michaelis–Menten light optimizing response curve (Beetz et al., 2013;Kandel et al., 2016). For each measurement location per measurement campaign, the GPP was modeled by the parameters $\alpha$ and $GPP_{max}$ (maximum photosynthetic rate with infinite PAR) of (eq.3):

$$GPP = \frac{\alpha \times PAR \times GPP_{max}}{GPP_{max} + \alpha \times PAR}$$

225                                                                          (3)

where GPP is the $CO_2$ flux measured with transperant chambers and corrected with $R_{eco}$, $\alpha$ is ecosystem quantum yield (mg $CO_2$ - C m$^{-2}$ h$^{-1}$)/($\mu$mol m$^{-2}$ s$^{-1}$) which is the linear change of GPP per change in PAR at low light intensities (<400 $\mu$mol m$^{-2}$ s$^{-1}$ as in (Falge et al., 2001), PAR is measured photosynthetic active radiation ($\mu$mol quantum m$^{-2}$ s$^{-1}$), $GPP_{max}$ is gross primary productivity at its optimum. Due to low coverage of the PAR range in a single measurement campaign, data from multiple

campaigns were pooled according to dates, vegetation, and air temperature.

### 2.4.4 Net ecosystem carbon balance calculations

The NEE is the sum of $R_{eco}$ and GPP values, calculated by applying the hourly monitored soil temperature (-5 cm) and PAR data to the models developed per campaign. Extrapolated values at times between two adjacent models are weighted averages of the estimates from these two models, where the weights are temporal distances of the extrapolated time spots to both of the

measurements. To account for the influence from plant biomass on the CO2 fluxes, linear relationships between grass height and model parameters ($R_{eco,Tref}$, $GPP_{max}$, and $\alpha$) were developed. Models developed for the campaign before harvesting were then corrected using the slopes of the linear regressions as the models after the harvest to be applied in the extrapolation. The loss of biomass was therefore accounted according to lowered grass height, different from the studies where model parameters are to zero after harvest (e.g., Beetz et al., 2013). Unrealistic parameters after correction were discarded, and instead adopted

from parameters from campaigns with low grass height at the same plot. The annual $CO_2$ fluxes were thus summing of the hourly $R_{eco}$, GPP and NEE values. The atmospheric sign convention was used for the calculation of NECB. All C fluxes into the ecosystem were defined as negative (uptake from the atmosphere into the ecosystem), and all C fluxes from the ecosystem

to the atmosphere are defined as positive. This also holds for non-atmospheric inputs like manure (negative) and outputs like harvests (positive). Both harvest and manure input are expected to be released as $CO_2$.

**245  2.4.5 CH$_4$ and N$_2$O fluxes**

$CH_4$ and $N_2O$ fluxes per site and measurement campaign were averaged per day. The annual emissions sums for $CH_4$ were estimated by linear interpolation between the single measurement dates. Global Warming Potential (GWP) of 34 t $CO_2$-eq for $CH_4$ was used according to IPCC standards (Myhre et al., 2013).

### 2.4.6 Uncertainties

The estimation of total uncertainties of the yearly budget should include multiple sources of error, where both model error and uncertainty from extrapolations in time are the most important (Beetz et al., 2013). Therefore, we included these two sources of error and combined them into a total uncertainty in three steps. First, we calculated the model error, which would cover the uncertainties from replications (between the three frames) and the random errors from the measurements, the environmental conditions at the time, and the parameter estimation of $R_{eco}$ and GPP. Standard errors (SE) of the prediction were calculated

for each measurement campaign / pooled dataset as the SEs of the midday of the campaign dates. The hourly SEs were then extrapolated linearly between modeled campaigns. Total model error of the annual NEE was therefore calculated following the law of error propagation as the square root of the sum of squared SEs. Second, we attribute the uncertainty from extrapolation to the variations from selecting different gap-filling strategies, since other approaches of annual NEE estimation including different $R_{eco}$ and GPP models would result in different values (Karki et al., 2019). To quantify this uncertainty, six

$R_{eco}$ models and four GPP models were select from Karki et al. (2019) and fitted with annual data (Appendix Table A1). The models were evaluated following the thresholds of performance indicators in Hoffmann et al. (2015). Fitted parameters of $R_{eco}$ and GPP models that performed above the 'satisfactory' rating were accepted and used to gap-fill NEEs. Based on all the annual NEEs per site and year, standard deviations from the means were considered as the extrapolation uncertainty. In the year 2018, the control site of farm D did not yield any satisfactory $R_{eco}$ model. The uncertainty was thus calculated as the

average of all sites. Finally, we calculated the total uncertainties per site and year following the law of error propagation with the uncertainties from the previous steps.

## 2.5 Statistics

The effect of the treatment on gap-filled annual $R_{eco}$ and GPP, the resulting NEE, the C-export data, the NECB, and the measured $CH_4$, $N_2O$ exchanges were tested by fitting linear mixed-effects models, with farm location as a random effect. Effectiveness of the random term was tested using the likelihood ratio test method. Significance of the fixed terms was tested via Satterthwaite's degrees of freedom method. General linear regression was used instead when the mixed effect model gives singular fit. The treatment effect was further tested using campaign-wise $R_{eco}$ data. Measured $R_{eco}$ fluxes from SSI and Control were calculated into daily averages and paired per date. The data pairs were grouped based on the GWT differences between SSI and control of the dates. Differences between treatments were then analyzed by linear regression of the $R_{eco}$ flux pairs without interception and testing the null hypothesis 'slope of the regression equals to 1'. All statistical analyses were computed using R version 3.5.3 (Team, 2019) using packages lme4 (Bates et al., 2014), lmerTest (Kuznetsova et al., 2017), sjstats (Lüdecke, 2019), and car (Fox and Weisberg, 2018).

## 3 Results

**3.1 Weather conditions**

Mean annual air temperature was 10.3 °C for 2017 and 10.7 °C for 2018, which were higher than the 30-year average of 10.1 °C. The growing season (April–September) in 2017 was slightly cooler with 14.3 °C than the average of 2018 at 14.6 °C, while the temperature during the growing season in 2018 was 1.1 °C warmer than average. Precipitation was slightly higher for 2017 840–951 mm compared to the 30-year average of 840 mm (KNMI). There was a small period of drought in May and June,

ending in the last week of June (See Fig.3). In contrast, 2018 was a dry year with average precipitation of 546–611 mm (range of two sites in Friesland). The year is characterized by a period of extreme drought in the summer, from June to the beginning of August, and precipitation lower than average in the fall and winter.

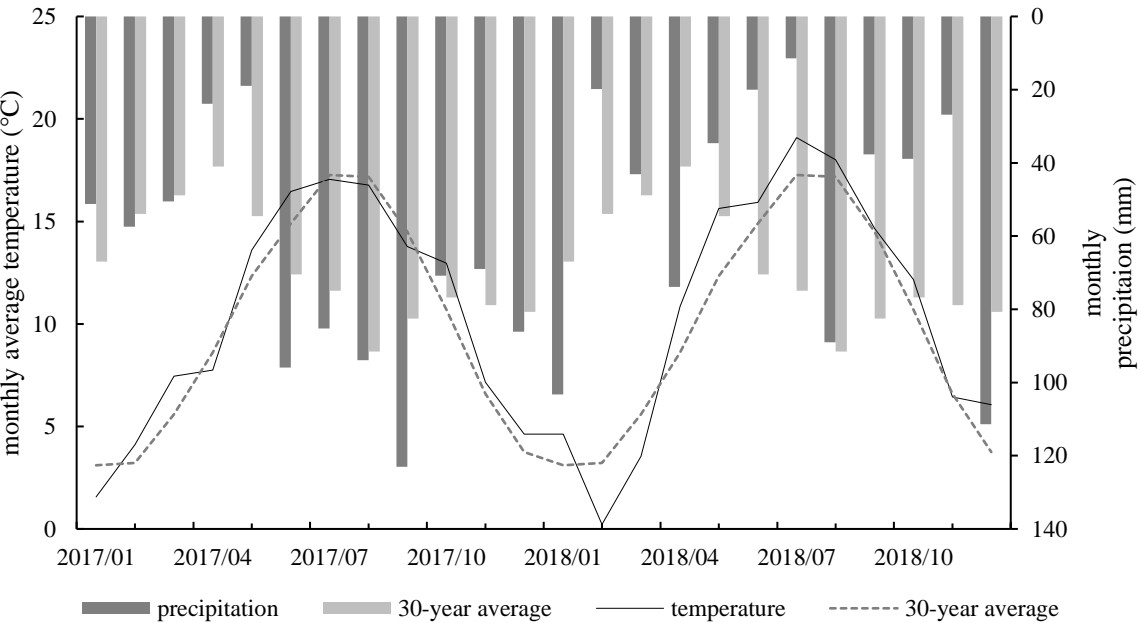

**Figure 3 Monthly average air temperature at weather station Leeuwarden (18 to 30 km distance from research sites), and the 30-year average. Sum monthly precipitation at weather station Leeuwarden, and the 30-year average.**

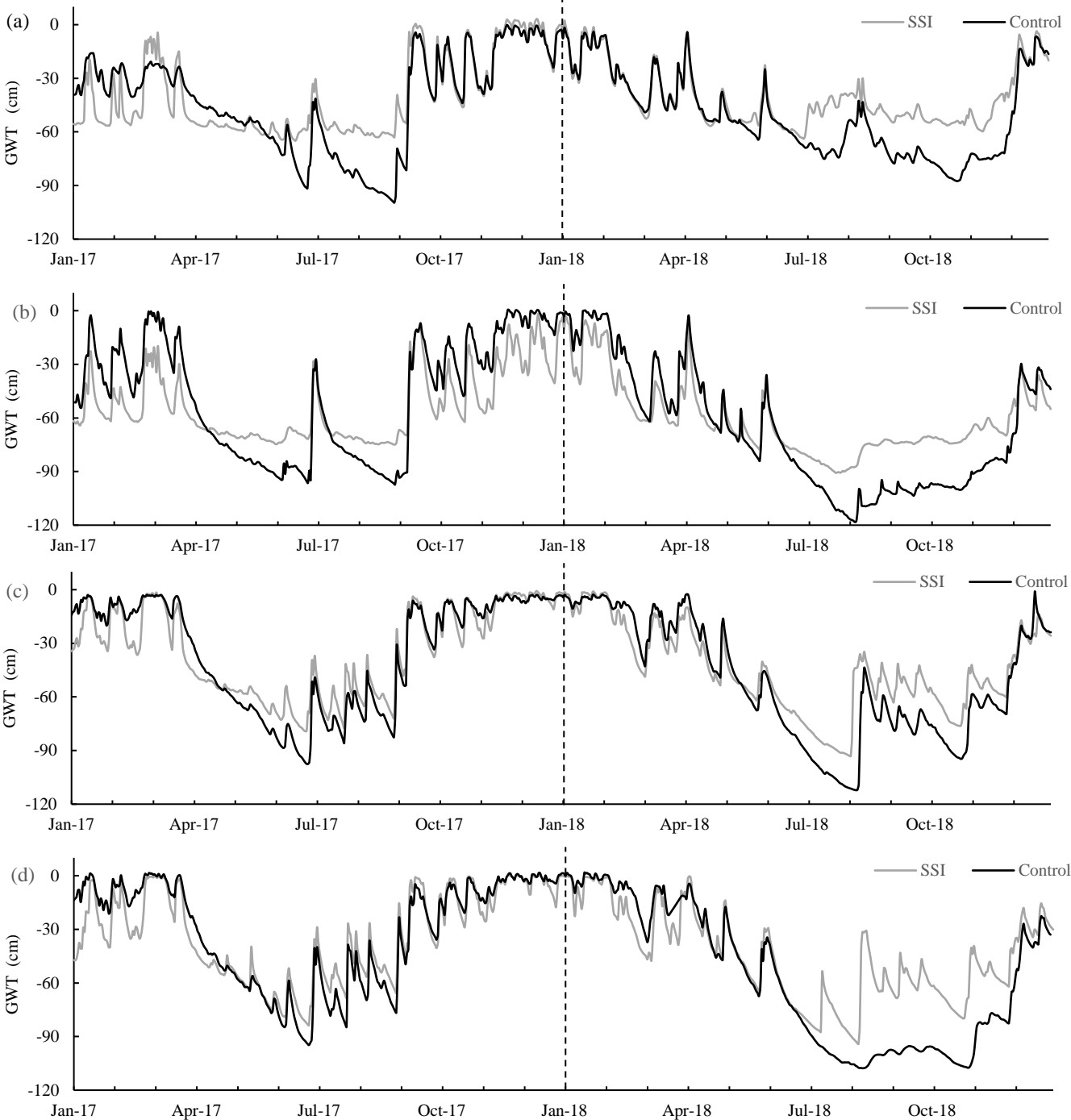

**Figure 4 Groundwater table (GWT, from soil surface) during the measuring period per farm (letter); values are shown for SSI (measured at 1.5 m from the irrigation pipe) and control treatments..**

## 3.2 Groundwater table (GWT)

Deploying SSI systems affected GWT dynamics during the two years for all farms (Fig. 4). However, there was a large variation in effect-size between years and locations. The effect of SSI can be divided into two types of periods. Periods with drainage (decreased GWT), in the wet periods, coincided with the autumn (in 2017) and winter period (2017 and 2018). Irrigation (increased GWT) periods, where the SSI leads to a higher water table than control, occurred during spring and summer when the GWT dipped below the ditch water level. In 2017, the effectiveness differed per farm. For locations A and B, GWT was more stable in summer around the -60 and -70 cm for SSI compared to the control, while locations C and D the GWT fluctuated more like in the control fields. During the dry summer of 2018, in contrast, all locations showed a strong effect of irrigation, especially after the dry period in the beginning of august. In this period the water table recovered quickly while the control lagged behind.

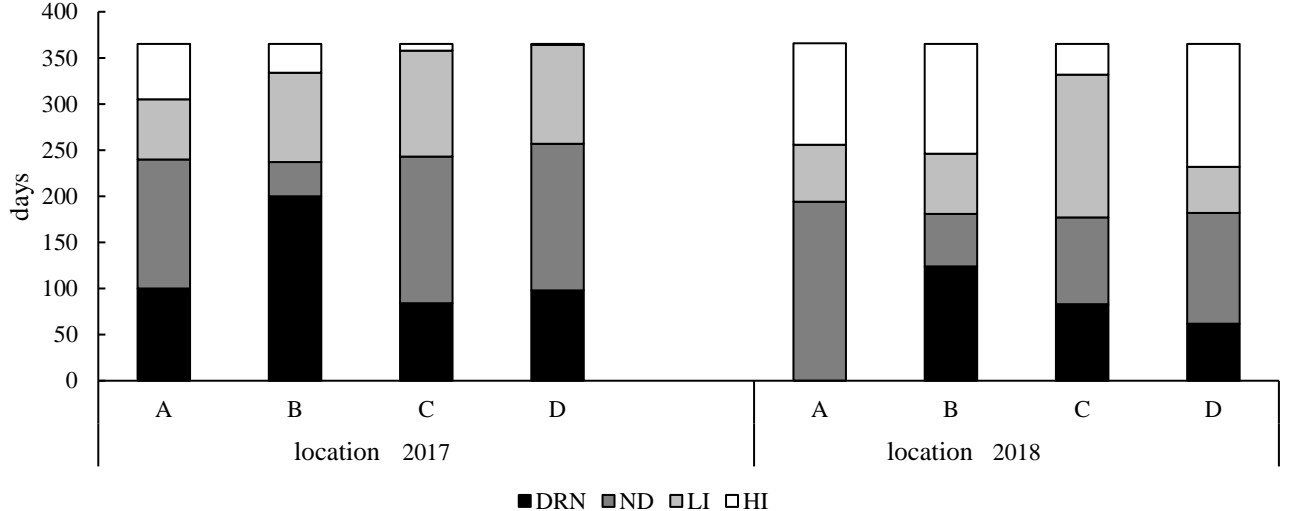

**Figure 5 Days with effective drainage/ irrigation for the four locations. drainage (DRN, <-5 cm), no difference (ND, -5 ~ 5 cm), low to intermediate irrigation (LI, 5 ~ 20 cm) and high irrigation (HI, > 20 cm) 1.5 m from the SSI pipe.**

Although there was hardly any difference in annual average GWT between control and SSI (< 5 cm; Table 2), drainage and irrigation effects could be observed when dividing the calendar year into seasons. The effective days of the SSI are summarized in Fig. 5 according to four classes, based on practical definitions of drainage and irrigation: drainage (DRN, <-5 cm), no difference (ND, -5 ~ 5 cm), low to intermediate irrigation (LI, 5 ~ 20 cm) and high irrigation (HI, > 20 cm). These classes are

also used in the statistical analysis of $R_{eco}$ measurements (see 3.7 Seasonal $R_{eco}$). In 2017 there were 17 days more without any GWT difference than in 2018. There was a much stronger irrigation effect in the dry year of 2018, with 61 more irrigated days comparing to 2017, and the number of irrigation days was constantly similar to, or higher than, the number of drainage days, except for site B in 2017 which had a long period showing a drainage effect.

**Table 2: Average Groundwater table (cm from the surface level) during the measuring period per farm. Summer groundwater table ranges from April till October. Measured 1.5 meter from the SSI pipe.**

| Location | Treatment | Average 2017 | Summer 2017 | Average 2018 | Summer 2018 |
|----------|-----------|--------------|-------------|--------------|-------------|
| A | SSI | -43 | -52 | -51 | -48 |
| | Control | -40 | -63 | -41 | -59 |
| B | SSI | -47 | -64 | -67 | -71 |
| | Control | -53 | -73 | -61 | -83 |
| C | SSI | -35 | -54 | -51 | -56 |
| | Control | -34 | -61 | -45 | -67 |
| D | SSI | -31 | -51 | -59 | -56 |
| | Control | -32 | -56 | -45 | -77 |

### 3.3 Measured $R_{eco}$

Figure 6 compares the measured $R_{eco}$ fluxes with the corresponding GWT measurements, which could give an indication for the effectiveness of the GWT differences. The classes were based on the GWT differences between the SSI and control sites on the measurement days (the same classes used in Fig. 5). There was a slightly higher $R_{eco}$ for SSI during drainage periods when GWT was lower (DRN), which compensates for the lower $R_{eco}$ during summer. For moments where there was no GWT difference (ND) and those showing moderate irrigation (LI), there was no effect of SSI on $R_{eco}$. However, when the GWT of the SSI was more than 20 cm higher than the control (HI), the emissions of the control were significantly higher than SSI ($p < 0.01$), indicating an effect of the irrigation. However, this effect of the raised GWT was small, even though in some cases the GWT was raised more than 60 cm. According to Fig. 5, in 2017, majority of the days were dominated by drainage (increasing $R_{eco}$), or by no difference or small irrigation resulting in no effect on the $R_{eco}$. However, periods with increased irrigation (Fig. 5), when there was a reduced $R_{eco}$ effect of SSI, were sparse compared to the other dominating periods.

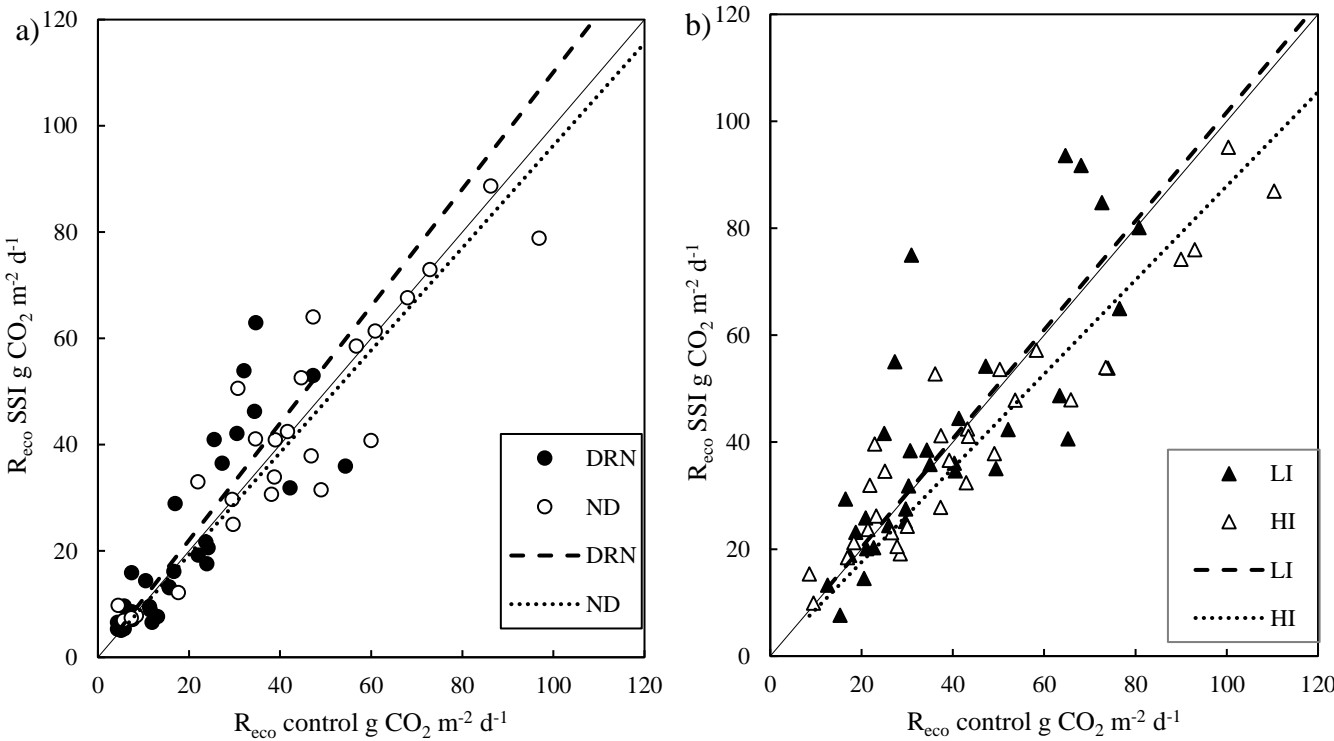

**Figure 6 Measured fluxes for ecosystem respiration ($R_{eco}$), one-to-one comparison in which daily averages were used. a) Values divided into two groups: with lowered groundwater table due to the effect of drainage (DRN), and with a small difference (ND). B) Values divided into two groups with irrigation effects, moderate infiltration with more than 5–20 cm difference (LI) and high infiltration (HI) with more than 20 cm difference between SSI and Control. Black filled line is the 1:1 line.**

**3.4 Annual carbon fluxes**

**3.4.1 Gross primary production (GPP)**

GPP was high for all locations in both years, showing a clear seasonal pattern with the highest uptake at the start of the summer (Fig.7). GPP was 30% lower in the dry year 2018 ($p < 0.001$) compared to 2017 (see Table 2) and differed between locations (random effect $p = 0.006$). There was, however, no treatment effect on GPP ($p = 0.3101$). Average GPP values for all SSI and

340 control plots were $-88.3\pm7.5$ and $-89.2\pm13$ t $CO_2$ ha$^{-1}$ yr$^{-1}$ for 2017, $-71.7\pm6.6$ and $-65.7\pm4.9$ t $CO_2$ ha$^{-1}$ yr$^{-1}$ for 2018, respectively.

### 3.4.2 Ecosystem respiration ($R_{eco}$)

$R_{eco}$ was generally high for all the farms measured during the two years, with the average $R_{eco}$ of 128.4±4.6 t $CO_2$ ha$^{-1}$ yr$^{-1}$ for 2017 being significantly higher than 100.8±11 t $CO_2$ ha$^{-1}$ yr$^{-1}$ for 2018 ($p < 0.001$) (Table 2). Different seasonal patterns were also observed between the two years, where in 2017 $R_{eco}$ peaked in June and July, while in 2018 the highest $R_{eco}$ was found in May (Fig. 7, Appendix B). However, no effect of SSI on $R_{eco}$ was found ($p = 0.6191$), with average $R_{eco}$ values for all SSI and control plots as 128.7±9.2 and 126.7±9.5 t $CO_2$ ha$^{-1}$ yr$^{-1}$ in 2017, 102.1±14.1 and 99.6±13.5 t $CO_2$ ha$^{-1}$ yr$^{-1}$ in 2018.

### 3.4.3 Net ecosystem exchange (NEE)

All locations functioned as large C sources during the measurement period. The average annual NEE of all sites amounted to 39.7±11 and 31.8±8.4 t $CO_2$ ha$^{-1}$ yr$^{-1}$ in 2017 and 2018, respectively. The overall explanatory power of year, treatment and location was low, with no yearly difference between 2017 and 2018 ($p = 0.1813$), or any treatment effect of SSI ($p = 0.9805$). The average NEE values for all SSI and control plots are 40.4±11.9 and 37.5±16.1 t $CO_2$ ha$^{-1}$ yr$^{-1}$ in 2017, 30.4±15.6 and 34±14.5 t $CO_2$ ha$^{-1}$ yr$^{-1}$ in 2018, respectively.

### 3.4.4 C-export (yield)

C-exports (i.e., yields) differed between years without treatment effect of SSI ($p = 0.691$). Following the drought in 2018, C export (13.8±0.6 t $CO_2$ ha$^{-1}$ yr$^{-1}$) was significantly lower ($p < 0.001$) than in 2017 (18.0±1.4 t $CO_2$ ha$^{-1}$ yr$^{-1}$). These values corresponded to dry matter yields of 9.4±0.6 t DM ha$^{-1}$ yr$^{-1}$ in 2018 and 12.6±1.1 t DM ha$^{-1}$ yr$^{-1}$ in 2017. The year-effect differed per location (random effect $p < 0.001$). We found a solid relationship between C-export and GPP ($p < 0.001$, $r^2 = 0.942$; linear-mixed modeling).

### 3.4.5 Net ecosystem carbon balance (NECB)

All sites are large carbon sources, without an effect of SSI ($p = 0.9446$) which was consistent for all farms (Table 3). However, there was a significant difference between the two years, with higher C emission rates in 2017 amounting to 49.6±11 t $CO_2$ eq. ha$^{-1}$ yr$^{-1}$ on average, compared with 36.9±7.6 t $CO_2$ eq. ha$^{-1}$ yr$^{-1}$ for 2018 ($p=0.0277$).

### 3.5 Methane exchange

The total exchange of $CH_4$ was very low during both years with no effect from the SSI (p=0.1147) or difference between years (p=0.1253). During most periods, the locations functioned as a sink of $CH_4$. The annual fluxes were $-0.01\pm0.01$ t $CO_2$ eq. ha$^{-1}$ yr$^{-1}$ ($-0.25$ kg $CH_4$ ha$^{-1}$ yr$^{-1}$) for 2017 and $-0.06\pm0.05$ t $CO_2$ eq. ha$^{-1}$ yr$^{-1}$ ($-1.8$ kg $CH_4$ ha$^{-1}$ yr$^{-1}$) for 2018 (Table 4). Such exchange did not play a significant part in the total GHG emissions (comparable to less than 0.4% of the annual NECB).

### 3.6 Nitrous oxide exchange

There was no treatment effect (p=0.5640) or inter-annual difference (p=0.4414) detected. The highest average emissions were measured on the SSI plot of location D, with $5.78\pm5.9$ mg $N_2O$. m$^{-2}$ d$^{-1}$ for 2017 and $10.7\pm17.4$ mg $N_2O$. m$^{-2}$ d$^{-1}$ for 2018. The highest peak was measured on the frame closest to the SSI pipe in August for SSI of location D, showing $55\pm15$ mg $N_2O$ m$^{-2}$ d$^{-1}$. The peaks observed were erratic and did not correspond to fertilization management with slurry before measurement campaigns.

**Table 3 Overview of all processes contributing to the carbon balance calculated for both years. Ecosystem respiration ($R_{eco}$), gross primary production (GPP), net ecosystems exchange (NEE, sum of GPP and $R_{eco}$), C-exports (harvest), C-manure (carbon addition from manure application), and net ecosystem carbon balance (NECB, sum of all fluxes) for subsoil irrigation (SSI) and control plots at farm locations A-D. The range of $R_{eco}$, GPP and NEE represent the combination of model error and extrapolation uncertainties following the law of error propagation.**

| | | | Carbon exchange | | | | | NECB |
|------|----------|-----------|-----------|-----------|-----------|-----------|-----------|-----------|
| Year | Location | Treatment | $R_{eco}$ <br> t $CO_2$ ha$^{-1}$ yr$^{-1}$ | GPP <br> t $CO_2$ ha$^{-1}$ yr$^{-1}$ | NEE <br> t $CO_2$ ha$^{-1}$ yr$^{-1}$ | C-export <br> t $CO_2$ ha$^{-1}$ yr$^{-1}$ | C-manure <br> t $CO_2$ ha$^{-1}$ yr$^{-1}$ | $CO_2$ <br> t $CO_2$ ha$^{-1}$ yr$^{-1}$ |
| 2017 | A | SSI | 125.9±3.4 | -88.8±2.7 | 37.1±4.4 | 16.6±0.4 | -6.9±0.1 | 46.8±4.4 |
| | | Control | 134.8±6.5 | -81.5±7.9 | 53.3±10.2 | 19.3±0.7 | -6.9±0.1 | 65.7±10.2 |
| | B | SSI | 125.2±5.8 | -97.8±3 | 27.4±6.5 | 15.3±1.1 | -5.3±0.1 | 37.4±6.6 |
| | | Control | 123.4±5.8 | -92.2±2.9 | 31.2±6.5 | 15.5±0.0 | -5.3±0.1 | 41.4±6.5 |
| | C | SSI | 132.5±4.6 | -87.9±5.7 | 44.6±7.4 | 22.1±0.2 | -10.9±0.2 | 55.8±7.4 |
| | | Control | 122.7±3.2 | 100.1±8.3 | 22.6±8.9 | 23.3±0.9 | -10.9±0.2 | 35±8.9 |
| | D | SSI | 134.6±4.2 | -78.6±2.8 | 56±5 | 15.7±1.4 | -9.3±0.2 | 62.4±5.2 |
| | | Control | 127.9±2 | -82.7±5.3 | 45.2±5.6 | 16.3±0.6 | -9.3±0.2 | 52.2±5.6 |
| 2018 | A | SSI | 98±6.5 | -74.9±2.5 | 23.1±7 | 14±0.0 | -7.4±0.1 | 29.7±7 |
| | | Control | 101.1±5.5 | -69.3±3.1 | 31.9±6.4 | 14±0.0 | -7.4±0.1 | 38.5±6.4 |
| | B | SSI | 118.1±10.1 | -73.8±3.4 | 44.3±10.7 | 13.8±0.6 | -9.3±0.2 | 48.8±10.7 |
| | | Control | 111.5±10.5 | -64.6±2.8 | 46.9±10.9 | 12.2±1.2 | -9.3±0.2 | 49.8±11 |
| | C | SSI | 109.2±5.8 | -83±4.6 | 26.2±7.4 | 15.7±1.0 | -9.3±0.2 | 32.6±7.5 |
| | | Control | 99.2±1.3 | -74.2±0.6 | 25±1.5 | 15.8±0.4 | -9.3±0.2 | 31.5±1.6 |
| | D | SSI | 82.9±4.5 | -56±2.2 | 26.9±5 | 13.4±0.23 | -9.3±0.2 | 31±5 |
| | | Control | 86.5±6.3 | -55.9±2.4 | 30.6±7 | 12±0.32 | -9.3±0.2 | 33.3±7 |


**Table 4 The average measured CH₄ and N₂O emissions subsoil irrigation (SSI) and controls for the four locations (A-D) for both years in mg m⁻² d⁻¹. The total CH₄ balance in CO₂ equivalents, using radiative forcing factors of 34 for CH₄ according to IPCC standards (Myhre et al., 2013). The ranges of CH₄ and N₂O represent the standard deviation (SD) of the measured fluxes.**

| | | | GHG fluxes | | Balance |
|---|---|---|---|---|---|
| Year | Location | Treatment | $CH_4$ | $N_2O$ | $CH_4$ |
| | | | mg $CH_4$ m⁻² d⁻¹ | mg $N_2O$ m⁻² d⁻¹ | t $CO_2$ eq. ha⁻¹ yr⁻¹ |
| 2017 | A | SSI | -0.44±0.5 | 0.02±0.7 | -0.01 |
| | | Control | -0.54±0.9 | 1.46±1.8 | -0.05 |
| | B | SSI | -0.43±0.4 | 3.81±3.3 | -0.04 |
| | | Control | -0.27±0.9 | 2.30±4.9 | -0.02 |
| | C | SSI | -0.43±1.0 | 2.48±1.5 | -0.03 |
| | | Control | -0.40±0.5 | 2.56±2.0 | 0.01 |
| | D | SSI | -0.50±0.8 | 5.78±5.9 | 0.01 |
| | | Control | 0.72±2.7 | 4.81±2.3 | 0.06 |
| 2018 | A | SSI | -0.39±0.7 | 0.15±0.8 | -0.05 |
| | | Control | -0.67±1.2 | 0.80±0.9 | -0.12 |
| | B | SSI | -0.40±0.3 | 2.08±3.7 | -0.04 |
| | | Control | -0.30±0.9 | 4.88±3.9 | 0.00 |
| | C | SSI | -0.73±0.9 | 3.27±3.0 | -0.11 |
| | | Control | -0.66±0.9 | 4.46±3.7 | -0.07 |
| | D | SSI | -0.91±0.6 | 10.7±17.4 | -0.09 |
| | | Control | -0.14±0.8 | 2.69±2.2 | 0.02 |


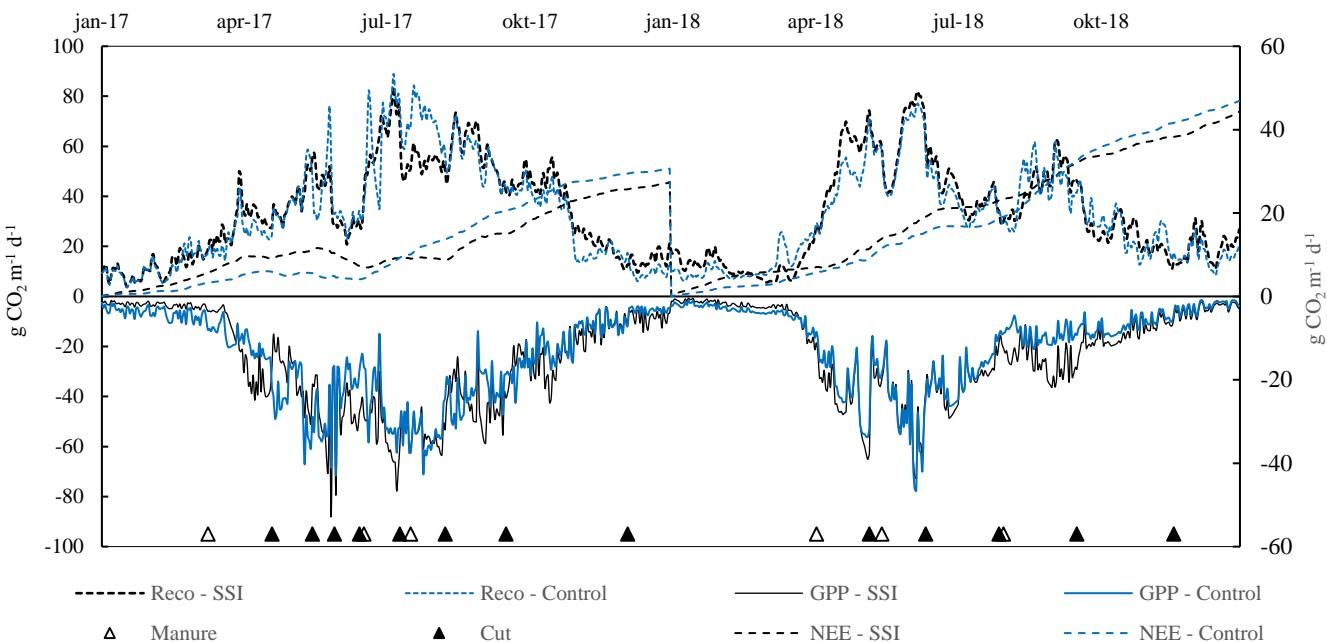

**Figure 7 R_eco and GPP for location B in g CO_2 m^{-2} d^{-1} on the primary y-axis, for control and SSI. Accumulative NEE in t CO_2 ha^{-1} yr^{-1}, for control and subsoil irrigation (SSI), every year starting at 0.**

## 4 Discussion

In this experimental research we found effects of subsoil irrigation (SSI) on water table dynamics without changing carbon dynamics profoundly. For both years, SSI had a clear irrigation effect during summer, increasing the averages of GWT during summer period by 6–18 cm at the four farms. During winter, there was a moderate but consistent drainage effect, reducing the average GWT in the wet/winter period by 1–20 cm. Mean annual GWT was little affected by SSI. Despite the irrigation effects and higher water tables in summer, there was no effect of SSI on $R_{eco}$, GPP and NEE in neither of the two years. We found no evidence for a reduction of $CO_2$ emissions, nor for yield improvements, on an annual base by implementing SSI.

### 4.1 SSI does not reduce annual $R_{eco}$

We identified three conditions that can explain the limited effect of SSI on carbon fluxes from the most prominent peat decomposition processes. Firstly, the uppermost 30–40 cm of the soil remains drained in both treatments throughout large

parts of the year (220–255 days) facilitating high $CO_2$ fluxes. Secondly, gas exchange from lower soil layers (60 cm and below) was presumably low due to moisture levels close to saturation that limit diffusion of $CO_2$ and $O_2$ effectively. Thirdly, the deliberate increase in drainage in the SSI treatment frustrate the irrigation effect on GWT. As a consequence, mean annual

GWT was similar for both treatments.

Based on the direct comparison using measured $R_{eco}$ fluxes (Fig. 6), we found a modest 5–10% reduction in $R_{eco}$ only when GWT differences were larger than 20 cm. When the irrigation effect was smaller, no effect on the $R_{eco}$ was found. An earlier study on intensively managed peat pastures in the Netherlands on the role of GWT also showed small effects of higher summer GWT on annual $R_{eco}$ and NEE despite substantial differences in soil volume changes/soil subsidence (Dirks et al., 2000).

Similarly, a 4-year study (Schrier-Uijl et al., 2014) found little differences in NEE estimates despite substantial variations in summer GWT and soil moisture contents.

It is generally assumed that higher GWT (mean annual or actual) leads to lower $CO_2$ emissions according to laboratory data (Moore and Dalva, 1993) and correlations between annual $CO_2$ fluxes and mean annual GWT (Wilson et al., 2016;Tiemeyer et al., 2020). However, there are also studies that did not find an effect of GWT on $CO_2$ emissions during the growing season

(Lafleur et al., 2005;Nieveen et al., 2005;Parmentier et al., 2009). This lack of effect is explained by the fact that there is only a small difference in soil moisture values above the GWT. The lower $CO_2$ emissions reported with structurally elevated GWT are often concomitant with substantial differences in vegetation/land use that are adapted to the higher GWT(Beetz et al., 2013;Schrier-Uijl et al., 2014;Wilson et al., 2016), which could confound the effects of GWT change. In our study, SSI seems to have an effect of a similar magnitude trending towards higher emissions during periods with lower GWT at the SSI sites.

The small treatment effect on measured $R_{eco}$ (Fig. 6) in our study can most probably be explained by differences in peat oxidation rates along the soil profile. Some other studies suggest that the top 30–40 cm of the peat profile play an important role in C turnover rates in drained peatlands, due to more readily decomposable C sources and higher temperatures (Moore and Dalva, 1993;Lafleur et al., 2005;Karki et al., 2016;Säurich et al., 2019). This soil layer was, however, not affected by higher summer GWTs in our study. SSI even reduced the number of days (24–27 days) that the top 30–40 cm soil layer

remained saturated, mostly in the wet season. Moreover, Säurich et al. (2019) speculated that the highest $CO_2$ production in the top 10 cm is reached when GWTs are approximately 40 cm below the surface. As the infiltrating water will affect the soil

moisture content of these layers, it is possible that SSI could even facilitate rather than mitigate summer emissions by approaching the optimum for C mineralization more often.

In contrast to surface irrigation, where the topsoil is replenished with moisture, the SSI effect is limited to deeper parts of the peat soils, at -60–-100 cm depth. However, the role of this deeper layer as a prominent C source for emissions to the atmosphere is likely to be limited. $CO_2$ production and export from deeper layers is prevented by lower temperatures, limited $O_2$ intrusion, and the fact that water content of this layer is already close to saturation which is frustrating gas diffusion (Berglund and Berglund, 2011;Taggart et al., 2012;Säurich et al., 2019). This layer shows low levels of stronger electron acceptors such as $O_2$ and nitrate used for the microbial oxidation of organic compounds, and of labile organic matter (Fontaine et al., 2007;Leifeld et al., 2012). Visually, the layers at our sites deeper than 60 cm were less decomposed (yellow–brown color with plant macrofossils still visible) compared to the highly degraded peat in the uppermost 40 cm layer.

In our case, although $CO_2$ production in deeper peat layers could be lower due to saturation after SSI induced GWT elevation, this reduction may be compensated by the increased $CO_2$ production in the top 20–40 cm due to the higher moisture levels resulting from elevated water levels. In the dry year of 2018, large differences between GWT in SSI and Control sites of up to 20 cm was observed, with the lowest summer GWT as deep as -120 cm in the control sites. A maximized effect of SSI would be expected according to the assumption from the Dutch soil-carbon-water model, where the average lowest summer GWT (i.e., GLG 'gemiddeld laagste grondwaterstanden') is considered as the major control of $CO_2$ emissions (STOWA, 2020). The absent of SSI treatment effect in this case provides additional evidence that SSI contributes little if any to the mitigation of $CO_2$ emission from drained peatlands. Such understanding of the processes of $CO_2$ emissions in relation to soil profiles should be further investigated.

## 4.2 SSI effects on $CH_4$ and $N_2O$ emission

The magnitudes of measured $CH_4$ and $N_2O$ fluxes are substantially lower than $CO_2$ fluxes, which would thus lead to negligible contributions to the total GHG emissions in our case. Looking directly at the measured fluxes, no SSI effect was detected for neither $CH_4$ or $N_2O$. Findings of this experiment agree with the generally accepted idea that intensively drained peatlands have low levels of $CH_4$ emissions, and often these systems even function as a small $CH_4$ sink (Couwenberg et al.,

2011;Couwenberg and Fritz, 2012;Tiemeyer et al., 2016;Maljanen et al., 2010). Drainage ditches, in contrast, emitted $CH_4$ at

high rates (Kosten et al., 2018;Lovelock et al., 2019). In the current study, the average of all measured $N_2O$ fluxes was 3.3

mg $N_2O$ m$^{-2}$ d$^{-1}$ (12 kg $N_2O$ ha$^{-1}$ yr$^{-1}$), which falls within the range of annual $N_2O$ emissions from drained peatlands in

Northwest Europe (4–18 kg $N_2O$ ha$^{-1}$) (Leahy et al., 2004;Maljanen et al., 2010;Schrier-Uijl et al., 2014;Kandel et al., 2018).

Fertilization, temperature and water table fluctuations play major roles in the total $N_2O$ emission (Regina et al., 1999;Van

Beek et al., 2011;Poyda et al., 2016). The mechanisms of $N_2O$ production and consumption in organic soils are, however,

complex and there is high temporal and spatial variability as influenced by site conditions and management (Leppelt et al.,

2014;Taghizadeh-Toosi et al., 2019). It is well studied that periods with frost and thawing result in high $N_2O$ emissions

(Koponen and Martikainen, 2004). In this study, the low measurement frequency in both years does not allow annual

estimations of $N_2O$ with enough representation of peak $N_2O$ emission. However, SSI effect still cannot be expected

according to the direct comparison of measured fluxes.

**4.3 Reasonably high NEE**

In contrast to the expected function of the SSI technique based on land subsidence data, no effect has been found on either

promoting the yield/GPP nor reduction on NEE and other GHG emissions. Our NEE estimate averaging all sites and years at

35.8 (22.6 – 56.0) t $CO_2$ ha$^{-1}$ yr$^{-1}$ is at the higher end of the ranges reported for drained temperate peatlands (Wilson et al.,

2016). Tiemeyer et al. (2020) reported 30.4 (5.1 – 40.3) t $CO_2$ ha$^{-1}$ yr$^{-1}$ for drained organic soils in Germany. In a Dutch case

study authors found a NECB of 20.1 t $CO_2$ ha$^{-1}$ yr$^{-1}$ average over the years 2005-2008 (Schrier-Uijl et al., 2014). Comparing

GPP and $R_{eco}$ estimates with earlier reports we find that  GPP of the sites was higher than values found by Tiemeyer et al.

(2016) for productive and drained peatlands (-70 ± 18 t $CO_2$ ha$^{-1}$ yr$^{-1}$) especially in the year 2017 (-88.7±7.2 t $CO_2$ ha$^{-1}$ yr$^{-1}$),

and falls back to the range in 2018 (-69.0±8.9 t $CO_2$ ha$^{-1}$ yr$^{-1}$) due to the drought induced decline of $CO_2$ uptake (Fu et al.,

2020). Higher GPP estimates seem reasonable given the high C-export in 2017 (on average 18.0 t $CO_2$ ha$^{-1}$) that was

substantially larger than the 8.5 t $CO_2$ ha$^{-1}$ reported by Tiemeyer et al. (2016) for grassland on organic soils. On the other hand,

the $R_{eco}$ values of the sites (128.4±4.6 and 100.8±11 t $CO_2$ ha$^{-1}$ yr$^{-1}$ in 2017 and 2018, respectively) are also at the higher end

of the range (97 ± 33 t $CO_2$ ha$^{-1}$ yr$^{-1}$ in Tiemeyer et al. (2016)). Extrapolation bias was excluded as a possible reason for this

high $CO_2$ emission, since testing of different $R_{eco}$ modeling approaches (including different model selection, data clustering procedure and removal of raw data outliers) did not yield substantially different $R_{eco}$ values.

High $R_{eco}$ values may partly be explained by the timing of our measurements. Järveoja et al. (2020) reported in a boreal natural peatland strong diel patterns of $R_{eco}$ with peaks at both midnight and midday. The authors show that daily carbon fluxes were
overestimated when models were developed including peak emission. In case a similar pattern of $R_{eco}$ applies to temperate highly productive and drained peatlands, the flux measurements with opaque chambers to estimate $R_{eco}$ would need to be spread more evenly during day (and ideally throughout the night). In our case, the flux measurements were unevenly distributed and concentrated around midday, which may have led to overestimation of $R_{eco}$ and, therefore, NEE overestimation. Assuming a structural overestimation of $R_{eco}$ by 15% results in lower NECB estimates (26 t $CO_2$ ha$^{-1}$ yr$^{-1}$) over all sites and both years.


Besides general methodological limitations of the close-chamber method, there are also a number of biochemical mechanisms that may explain the high emissions found here. Abiotic conditions that favor high $CO_2$ emissions were present, with high temperatures for both years and non-limiting moisture conditions for 2017. Research from Pohl et al. (2015) found in a drained peatland a high impact of dynamic soil organic carbon (SOC) and N stocks in the aerobic zone on $CO_2$ fluxes. In our case, the
peat soils contained a high amount of C, especially in the upper 20 cm layer. This layer was also aerobic for long periods during the experiment, thus promoting high rates of C sequestration and decomposition. In conclusion, NEE estimates in the current study are high owing to systemic overestimation of $R_{eco}$ and conditions promoting high soil $CO_2$ production and release.

**4.4 Uncertainties**

GHG emissions from peat grasslands are highly variable (Tiemeyer et al., 2016) given the uncertainties from the wide ranges
of land use and management activities (Renou-Wilson et al., 2016) and gap filling techniques (Huth et al., 2017). In this study, besides the model errors inherent in the model development process, uncertainties from gap-filling techniques in terms of data-pooling strategies and model selections were also considered. Campaign-wise fitting of $R_{eco}$ and GPP models can best represent the original data sets, while pooling data for a longer period can provide better model fitness and less bias toward single

measurements (Huth et al., 2017;Poyda et al., 2017). However, in this study, different responses of vegetation and soil

processes to drought, especially to the extreme drought in 2018, caused data points that could not be explained by the classic

models, resulting in the generally poor performances of annual models. For this reason, we reported the annual budgets with

campaign-wise gap-filled NEE values. The uncertainties of NEE estimates from model differences were on average 14 tons

and up to 25 tons of $CO_2$. Nevertheless, no SSI effect was found considering NEE estimates from annual models. The model

differences quantified here were in good agreements with other model tests (Görres et al., 2014;Karki et al., 2019) and match

the magnitude of NEE uncertainties calculated with other methods (e.g. the 23–30 tons $CO_2$ variances reported by (Schrier-

Uijl et al., 2014) using eddy co-variance techniques). Additionally, $CO_2$ fluxes and annual budgets derived from eddy co-

variance approach in 2019 at location A support findings of the present study (Van den Berg and Kruijt, 2020). The eddy co-

variance revealed virtual identical flux patterns for both control and SSI field despite drastic differences in summer GWT

surpassing 80 cm at the height of the vegetation period.

**4.5 The effects of SSI on land use**

The intensity of land use (intensity and timing of drainage and fertilization, plant species composition, mowing and grazing

regimes) is a major driver of carbon turnover in grasslands (Renou-Wilson et al., 2016;Smith, 2014;Ward et al., 2016). SSI

facilitates earlier fertilization compared to management under current drainage systems by increasing the load-bearing

capacity of the field surface for fertilizing equipment. We expect nutrient accumulation in the soil to continue that can lead

to high $CO_2$ losses accelerated by nitrogen or phosphorus (Tiemeyer et al., 2016;Säurich et al., 2019). It was expected that C-

export via crop yields due to extra drainage could increase in a wet autumn. However, we did not find any indication for an

increase in land-use intensity or yield as a result of SSI. In summary, land-use intensity will remain high in SSI treatments

without substantial changes to carbon sequestrating vegetation (e.g., (Couwenberg et al., 2011;Schrier-Uijl et al.,

2014;Tiemeyer et al., 2020), tillage (Smith, 2014) or fertilization (Pohl et al., 2015).


The implementation of SSI may further inflict high costs on land users. Next to investing in 1800 to 2500 m of extra drainage

pipes per hectare maintenance costs rise. Drainage pipe inspection, cleaning and maintenance cost range between 0.30 to 0.90

€ per m with an incurrence interval of 3–6 years depending on abiotic conditions (K. Kooistra, personal communication, 2020). SSI inflict practical challenges in all catchments where ditch water levels are difficult to control and where water needs to be pumped in during summer. Groundwater extraction has been suggested as an alternative which will further increase direct costs (pumping infrastructure, fuel) and indirect costs including land-subsidence following groundwater extraction (Herrera-García et al., 2021). A large roll out of SSI seems costly, impractical and holds only few benefits for land use on peatlands.

## 5 Main conclusions

The implementation of SSI technique with the current design does not lead to a reduction of GHG emissions from drained peat meadows, even though there was a clear increase in GWT during summer (especially in the dry year of 2018). We therefore conclude that the current use of SSI with the aim to raise the water table to -60 cm is ineffective as a mitigation measure to sufficiently lower peat oxidation rates and, therefore, also soil subsidence. Most likely, the largest part of the peat oxidation takes place in the top 40 cm of the soil, which remained drained. This layer is still exposed to higher temperatures, sufficient moisture, oxygen and alternative electron acceptors such as nitrate, and nutrient input. We expect that SSI may only be effective when the GWT can be raised permanently to water tables close to the soil surface.

**Data availability.** The data are available on request from the corresponding author (S.T.J. Weideveld).

**CRediT authorship contribution statement:**

SW: Investigation, Data curation, Writing – original draft, Visualization, Methodology. WL: Investigation, Data curation, Writing – original draft, Visualization. MB: Data curation, Writing – original draft, Visualization. LL: Writing - review & editing, Supervision. CF: Conceptualization, Methodology, Data curation, Writing - original draft, Supervision

**Acknowledgements**

We would like to thank all technical staff, students and others who helped in the field and in the laboratory, as well as the landowners who granted access to the measurement sites. We acknowledge Peter Cruijsen and Roy Peters for their assistance

in practical work and analyses. Grants: WL is supported by the China Scholarship Council. MB was supported by the NWO-Peatwise grant. CF received funding from ERA-NET Climate Smart Agriculture.

**Appendix A Annual models**

 Table A1. Model selected for annual-model gap-filling approach of year budgets (adopted from Karki et al. 2019), as a measure of extrapolation uncertainties.

| Model | | Structure | Description |
|---|---|---|---|
| $R_{eco}$ | 1 | $Reco_{T_{ref}} * e^{E_0*\left(\frac{1}{T_{ref}-T_0}-\frac{1}{T-T_0}\right)}$ | Arrhenius function used for the campaign-wise model fit. Parameters follow descriptions in Material and Methods. |
| | 2 | $(Reco_{T_{ref}} + (\alpha * GH)) * e^{E_0*\left(\frac{1}{T_{ref}-T_0}-\frac{1}{T-T_0}\right)}$ | Model 1 adding $GH$ (grass height) as a vegetation factor. $\alpha$ is a scaling parameter of $GH$. |
| | 3 | $Reco_{T_{ref}} * e^{E_0*\left(\frac{1}{T_{ref}-T_0}-\frac{1}{T-T_0}\right)} + (\alpha * GH)$ | Different form of vegetation included Model 1. |
| | 4 | $R_0 * e^{bT}$ | Exponential function. $R_0$ is respiration at 0 °C, $b$ is a temperature sensitivity parameter. |
| | 5 | $(R_0 + (\alpha * GH)) * e^{bT}$ | Model 4 with vegetation included. |
| | 6 | $R_0 + (b * T) + (\alpha * GH)$ | Linear function. |
| GPP | 1 | $\dfrac{\alpha * PAR * GPP_{max}}{GPP_{max} + \alpha * PAR}$ | Michaelis-Menten light response curve as used for the campaign-wise model fitting. |
| | 2 | $\dfrac{\alpha * PAR * GPP_{max} * GH}{GPP_{max} * GH + \alpha * PAR} * FT$ | Model 1 with vegetation and air temperature included. FT is a temperature dependent function of photosynthesis set to 0 below - 2 °C and 1 above 10 °C and |

| | | | |
|---|---|---|---|
| | | | with an exponential increase between - 2 and 10 °C. |
| | 3 | $$\frac{GPP_{max} * PAR}{\kappa + PAR} * \left(\frac{GH}{GH + a}\right)$$ | Another form of the Michaelis-Menten light response curve with a vegetation term included. $a$ is a model-specific parameter. |
| | 4 | $$\frac{GPP_{max} * PAR}{\kappa + PAR} * \left(\frac{GH}{GH + a}\right) * FT$$ | Model 3 with air temperature included. |

## Appendix B Reco ,GPP and NEE

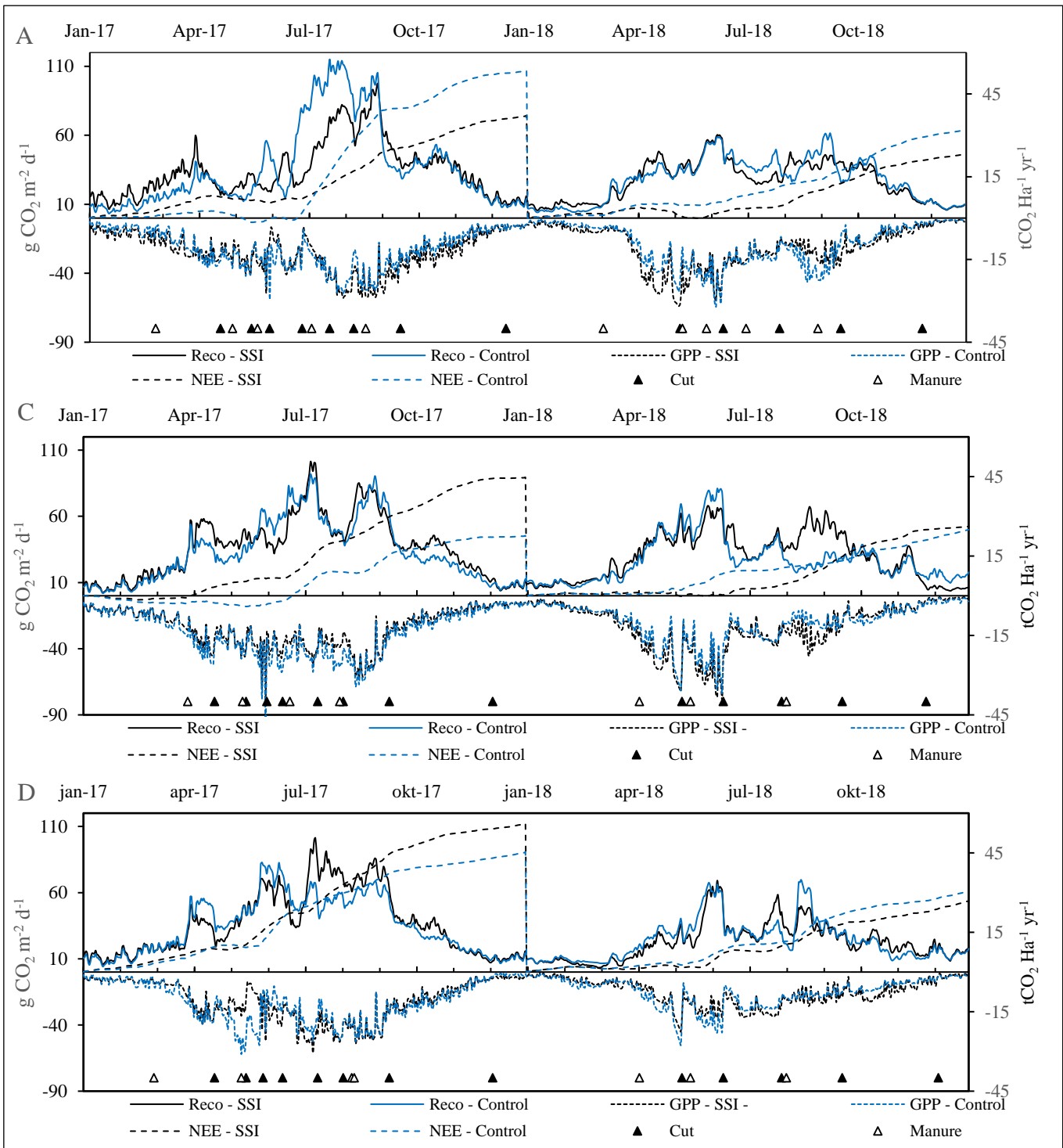

**Figure B1 Daily R_{eco} and GPP for location in g $CO_2$ $m^{-1}$ $d^{-1}$ on the primary y-axis, for control and SSI for locations A,C and D. Accumulative NEE in $tCO_2$ $Ha^{-1}$ $yr^{-1}$, for control and SSI, every year starting at 0.**

 **Appendix C CH₄ exchange**

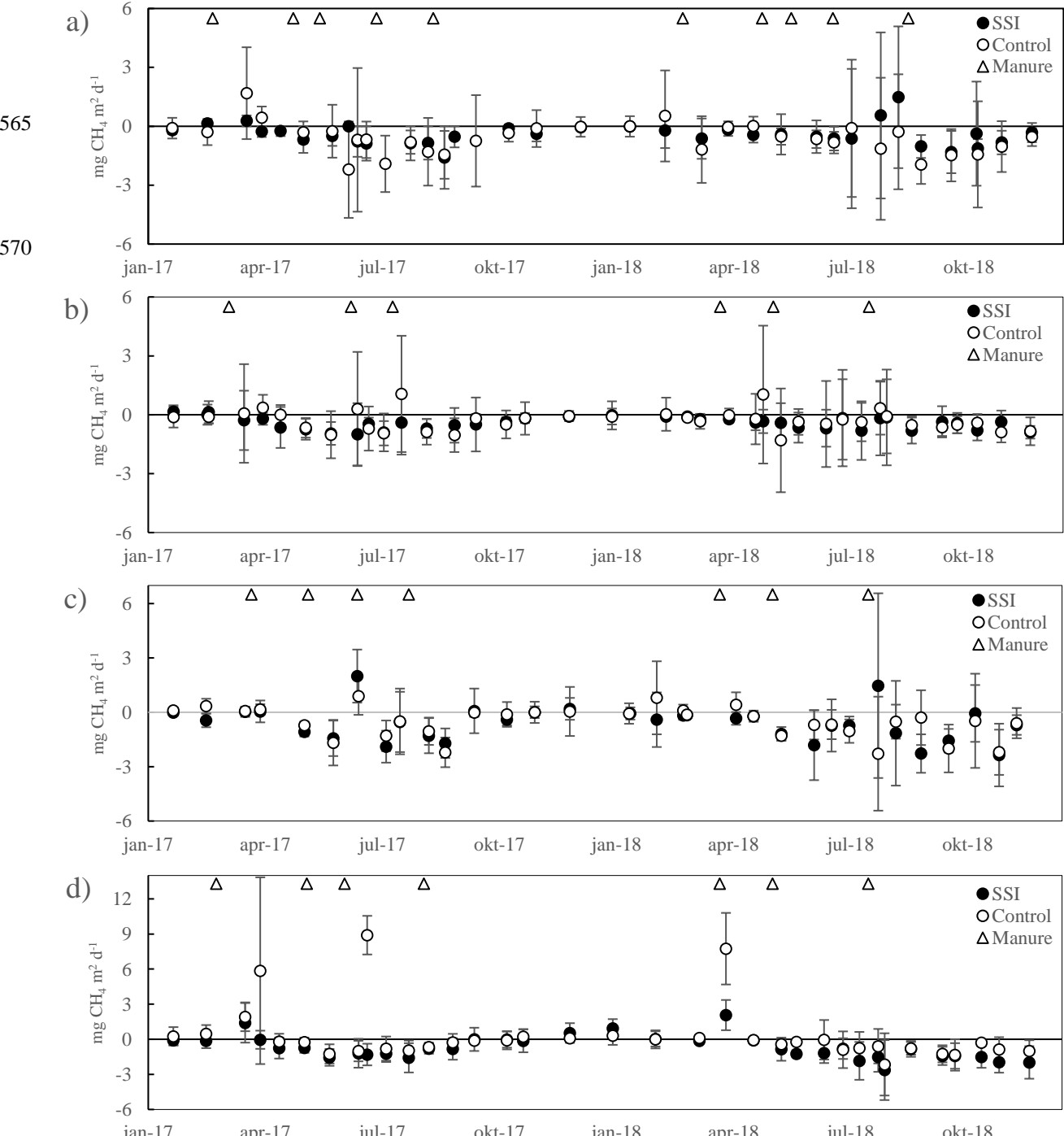

**Figure C1 CH₄ exchange throughout 2017 and 2018 in mg CH₄ m⁻² d⁻¹**

**Appendix D N₂O exchange**

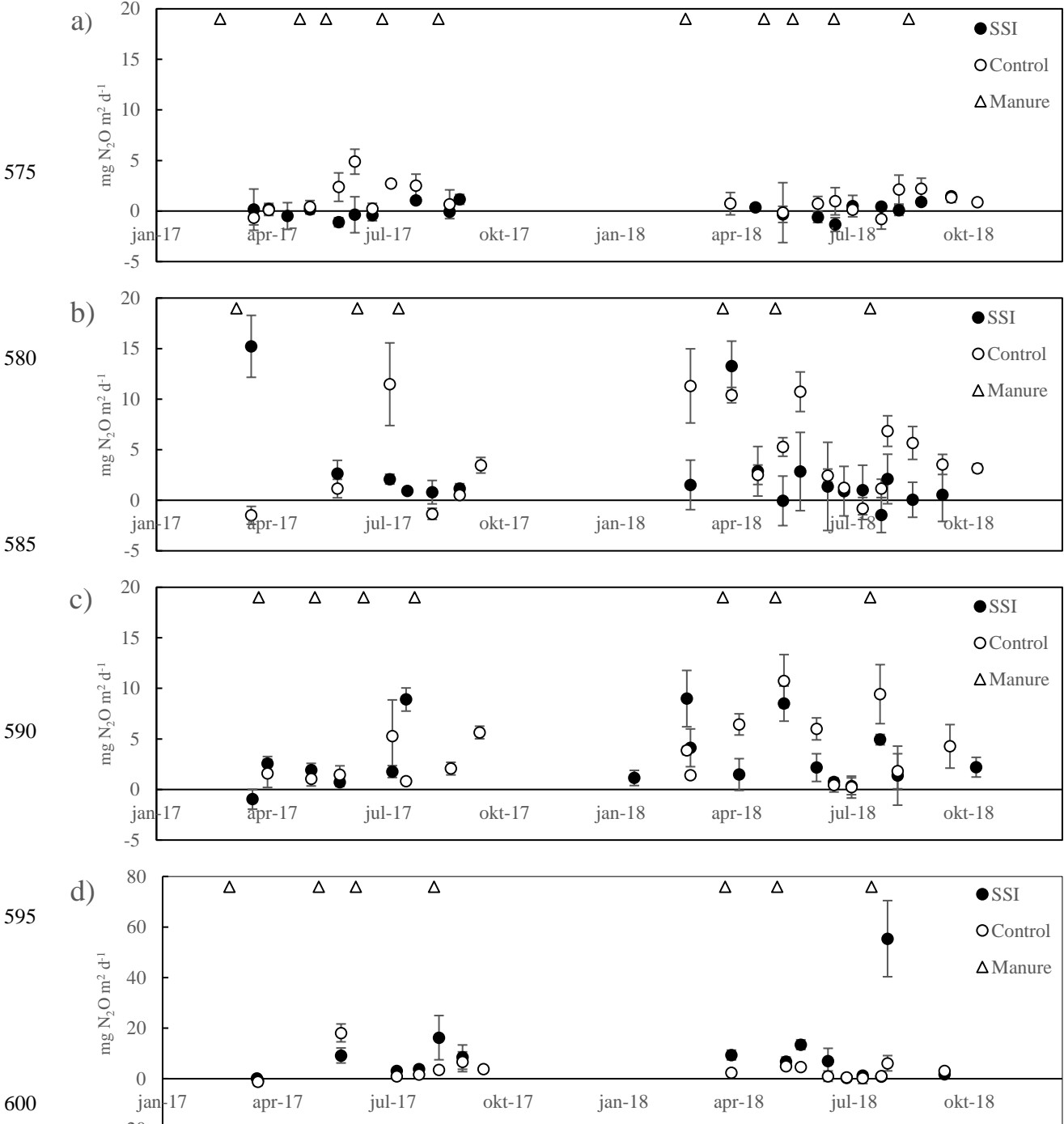

**Figure D1 N₂O exchange throughout 2017 and 2018 in mg N₂O m⁻² d⁻¹.**

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
