# Peer review of "Conventional sub-soil irrigation techniques do not lower greenhouse gas emission from drained peat meadows"

_Biogeosciences, 2020_

## Short Comment (SC1) · 20 Jul 2020

Dear authors,

Having been involved in the latest discussions on the research presented in this paper, I would like to post 2 comments:

1. In 2019, on some of these study sites, additional emission measurements have been made, using eddy covariance techniques. These measurements indicate far lower emissions as the ones with closed chambers, used in this study. It was recognised by the researchers that this might be due to an erroneous gap filling procedure. This

should be mentioned and explained in this paper.

2. The results do not indicate a difference in emissions rates between the sites with subsurface infiltration and the control sites. It should be mentioned that this conclusions is valid for the design of the infiltration and the soil type as used in these experiments, i.e. - drains at a depth of 60 cm and at a distance of 6 meter. - sphagnum peat type with a very low permeability. These additional remarks are required because, despite the obvious results for these study sites, it might be possible that subsurface drainage leads to lower emission rates when applied with alternative designs ( e.g. lower depth, smaller distances, in soils with higher permeability, with addional water reservoirs to increase water pressure etc. ). Consequently it also is important to adapt the title of the paper.

---

## Referee Comment (RC1) · Anonymous Referee #1 · 1 Aug 2020

General comments:

The authors investigate the GHG reduction potential of drained peatlands by using sub-soil irrigation. The topic of the paper is of relevance to Biogeosciences and will be of interest to an audience interested in mitigating greenhouse gas emissions from agriculturally used peatland. It is a novel approach, which needs further research. For the evaluation of the effect of sub soil irrigation on GHG emissions, a paired design of a control site and a sub-soil-irrigated site is used. Four different sites were investigated. $CO_2$, $CH_4$ and $N_2O$ fluxes were measured with chambers over a two 2 years period. Carbon and greenhouse gas budgets are determined and compared for the paired

sites.

I do not understand the experimental setup: The basic hypotheses of the manuscript that main GHG emissions comes from soil layers deeper than 70 cm is not well explained. Moreover, no information about soil properties and soil moisture of this relevant depth are given in the manuscript. As these soil data are missing, it is not clear, which amount of soil organic carbon is exposed to oxygen due to the alterated groundwater level. Often the bulk density is low in deeper peat layers. It would be interesting to calculate the additional % of aerated carbon due to the alteration of the groundwater level and to compare it to GHG emissions. Moreover, the authors should estimate if the small differences in groundwater level (<20 cm) can lead to a theoretical GHG reduction, which can be measured with this method (and associated uncertainties) and experimental set up. Unfortunately, soil moisture was only measured to soil depth of 20cm. (At site A, C, D, soil moisture and temperature is measured only in the mineral soil cover). Moreover, these data are not presented in the paper, although the importance of soil moisture is discussed in the discussion section. The main conclusion that SSI does not lower GHG emissions cannot be drawn from the presented data as most of time the difference in ground water level between the treatments was relatively small. However, when the differences in ground water level were > 20 cm a reduction of GHG emissions was observed. In my view, the conclusion from this paper would be that a substantial increase in groundwater level is needed to allow large enough effects in the emissions to be measured.

For two sites, the comparability of control and SSI treatments is not given. This may influence the mineralization processes and thus the results. In particular, Site A: SSI has considerably higher organic matter content (39 vs 27%) as the control, and C/N ratios (29 vs 20) indicate different organic matter quality. Moreover, Site D: Control site has nearly the double amount of organic matter than SSI (38 vs 61%). This aspect is not discussed in the manuscript and might bias the results.

In particular, the methods used to measure the carbon and greenhouse gas fluxes and

management are not described in sufficient detail.

The annual N2O budget was calculated based on only few measurement campaigns. In my opinion, it is not possible to calculate annual N2O budgets from 6-9 daily values, which were measured within 6 month in 2017. This is most evident at Site B Control: Linear interpolation of the high N2O emission in March probably overestimate the N2O emissions for the whole winter time. Moreover the material and methods section is misleadingly stating that N2O was measured for each measuring camping, but at Site B 38 campaigns were made, whereas I counted only 17 data points in Figure C1. Researchers reading the manuscript without looking at the supplementary data could extract the N2O data for annual budgets. Due to the low temporal resolution of the N2O data, it is not possible to distinguish between background N2O emissions and fertilizer-induced N2O emissions. As no daily data are presented for the CH4 budget, the data coverage and thus quality of annual budget cannot be evaluated.

The description of management is very short and important information about cutting days, fertilization events, and amount of applied fertilizer are missing, which makes it difficult to understand N2O and NEE data. E.g. Why are the cutting days not visible in the GPP data? In other studies, the decrease in GPP after management events can be nicely seen (Poyda et al. 2016 or Beetz et al 2013,). In comparison to their data, the GPP stayed rather constant and relatively high (-10 g CO2 m-2 d-1) throughout the year. Accordingly, it is difficult to evaluate the quality of GPP modeling.

The uncertainty assessment is nicely done for the gap filling method of NEE, but the uncertainty estimates are not integrated in the results and transferred to GPP and Reco. For NEE, Reco, and GPP an uncertainty is indicated, but it is not stated what it is (error of SD or 95% confidence interval . . .). The uncertainty range is given for NEE as 3-16 t CO2 ha-1 yr-1, (L 370), but NEE uncertainty from NEE gap filling is given as 14-25 t CO2 ha-1 yr-1). For N2O and CH4 the uncertainty assessment is missing. Other sources of uncertainty (systematic errors of the use of chamber methods or random errors) are not discussed. Please provide a more thorough uncertainty estimation of

all component of the net ecosystem carbon balance and included this values in Table 2 and 3.

Specific comments:

Experimental set up and management Table 1: please provide more information about soil properties of the mineral soil cover, and underlying peat layers (carbon content, bulk density, C/N and carbon stock) in a higher resolution for the entire aerated soil depth. Please state how many soil samples were taken per depth and where. Please also add the information of the depth location of the schalter. Figure 2: please provide information about the location of the chambers of the control site relative to the main ditch. Please provide data of cuttings days, fertilization events and measurement campaigns for CO2, N2O and CH4 as the growth of the grass and thus GPP strongly depend on time of measurement (days after cutting). Information can be added in Figure 7, Appendix B1, C1. Please add information about amount and determination of N und C input through slurry application. Please add information about the determination of the yield (dry mass of the grass). According to Table 1 Site A und B were grazed. How was carbon import through cattle manure determined? How was the carbon export through grazing determined? Was the yield only determined within the chambers or for the entire grassland? How was the grass height determined?

Gas fluxes Chambers: Was the location of the frames fixed over the two years? Did the vegetation change within the chambers during the experiment? Please add the transparency of the chambers? Was a correction term introduced due to a reduced transparency? Please add information about the used sign convention, positive fluxes = loss of carbon? Please add information about the used equation for the calculation of GHG balances and assumptions (harvest is assumed to be released as CO2?, loss of dissolved organic carbon?)

L242ff: what is the accuracy of precipitation data derived from satellite images?

L243: June 2017 seemed to have received more than the average precipitation, but

June is included in the drought period?

Figure 4: As there were 3 groundwater measurements per site, it is not clear which groundwater table is presented, average of all 3?, what is the SD of the three wells? How is the variability of groundwater level of the control site? Please explain DRN

L276: I do not understand the sentence "There is variation.." please clarify.

L327-330: What is meant by uncertainty of 3-16 t $CO_2$ ha-1 yr-1. What is represented by 1.6 t $CO_2$ ha-1 yr-1 I for NEE in 2017?

L326-332: What is the difference between annual NEE of 47 t $CO_2$ eq. ha-1 yr-1 (L327) and emissions of 62 t $CO_2$ eq. ha-1 yr-1 (L313)

L 334-338: Please provide daily $CH_4$ data.

Table 2 and Table 3: Please add uncertainty estimates for all components of GHG balance

L. 380: Reco was lower when the differences of groundwater level was >20 cm

L. 420: $N_2O$ emissions are not only driven by fertilization events, but also by soil moisture, which should be differ by the treatment. Thus, the comparison can be biased by missing peak events.

L428: please use the same sign convention for all cited references.

Technical comments: L310: Please state was the 4 t are, SD?, . . .,

Please indicate A und B in Figure 6

Figure 7: please use colors, which can be clearly distinguished

---

## Referee Comment (RC2) · Anonymous Referee #2 · 16 Aug 2020

Review

**Sub-soil irrigation does not lower greenhouse gas emission from drained peat meadows**

by Stefan Weideveld *et al*.

Generally, the manuscript will be of interest for readers of Biogeosciences, and the topic of adequate mitigation strategies for drained organic soils is one of high relevance. While the overall result that there is no difference in GHG emissions of this sub-surface irrigation (SSI) system and the control seems to be robust, there are, in my opinion, still four major issues which need to be solved before the manuscript could be considered for publication in BG:

- The authors appear to be surprised that SSI does not result in lowered GHG emissions, but this "surprise" is rather unfounded as the water table is raised only slightly towards a target level of 60 cm below ground, which I would – in line with the IPCC Wetlands Supplement (IPCC, 2014) – still regard as "deeply drained".

- This target water level seems to be based on the assumption that the majority of the $CO_2$ emissions originates from deeper peat, but no reasons are given for this assumption.

- According to Figure C1, there were partially only 7 measurement dates for $N_2O$ in 2017 and afterwards a gap of five months. Given the highly episodic nature of $N_2O$ fluxes, this is absolutely inadequate for the calculation of annual balances in a strongly fertilized grassland.

- For the interpolation of GPP, all measurement campaigns have been pooled for 2017 and harvests have not been accounted for when interpolating GPP despite the large influence of aboveground biomass on maximum photosynthetic rates.

**Title and assumption that this specific SSI system would lower GHG emissions**

The SSI system studied here has a target water level of -60 cm. Given the limited hydraulic conductivity of the peat and the "exit resistance" of the pipes, a water level of -60 cm in the ditches results in even deeper field water levels in summer. This target seems to be based on the assumption that $CO_2$ emissions originate from deeper peat (see below). Thus, the authors state that a WT rise of 6-18 cm in summer compared to an even lower level "unexpectedly" (line 22) or "contrary to our expectations" (line 29) does not lower GHG emissions. In my opinion, this is absolutely no surprise, but should be expected as laboratory studies often show highest respiration rates at medium water content and as field studies, on average, showed an asymptotic rather than a linear response of $CO_2$ emissions to water table depth (too dry, no more peat exposed, Tiemeyer *et al.*, 2020).

Thus, the title needs to be changed to "Sub-soil irrigation with target water levels of 60 cm does not lower carbon dioxide emissions from drained peat meadows" or something similar, as the experiments do not allow for conclusions on SSI in general. Further, if the authors are really surprised by their results, they will need to convince the reader *why*. In this context, it also needs to be discussed why such low target water levels have been chosen at all. At least for meadow use as in 2018, such low water levels are technically not needed when adequate machinery (low weight, double tyres, etc.) is used.

**Peat layers below -70 cm contribute most to GHG emissions**

In the introduction, there is no reasoning why this should be the case at all. Many studies have shown that topsoils show higher respiration rates than subsoils e.g. due to higher nutrient contents or generally more favourable conditions for microbial activity (e.g. Bader *et al.*, 2018). This is indeed briefly discussed on page 22, but the whole "story" of the manuscript (and probably also the design of the sub-surface irrigation system) builds on this assumption. Thus, either it needs to be substantiated by peer-reviewed (!) literature, or the manuscript needs to be restructured based on more adequate hypotheses.

**Frequency of $N_2O$ flux measurements**

According to Figure C1, there seem to be only 7 measurement dates for $N_2O$ in some cases in 2017, then a gap of more than 5 months in winter and finally a further gap of two months at the end of the study period. This contradicts the text that $N_2O$ was measured at each campaign, i.e. supposedly bi-weekly in summer and monthly in winter (page 8). If Figure C1 is actually correct, this data may not be used for the calculation of annual balances as effects of fertilisation cannot be captured adequately with such a low temporal resolution. Further, I would suspect that the first fertilisation event took place before April and was thus missed by the campaigns. In any case, fertilisation dates should be indicated in Figure C1.

Even more important, it is well-known that high $N_2O$ emissions may occur when temperatures change between frost and thaw (e.g. Koponen and Martikainen, 2004), especially under wetter conditions, and that maximum $N_2O$ fluxes of drained peatlands may occur in winter also under temperate climatic conditions (e.g. Flessa *et al.*, 1988). Therefore, the authors should refrain from calculating annual balances from a dataset without winter data. The $N_2O$ data could, however, be used to compare treatment effects on the basis of campaigns. In consequence, this means that GHG balances cannot be calculated from the presented data, but only C balances.

**GPP modelling**

In my opinion, pooling all summer data as done for 2017 is not an adequate gap-filling strategy as $GPP_{max}$ and $\alpha$ strongly depend on vegetation development. This strategy of pooling might be valid for (semi-)natural vegetation, but no for intensively used grasslands with frequent harvests. Further, it seems that parameters are generally interpolated across harvests which does not capture the effects on GPP, which should be very low after harvests. Harvests are unfortunately not indicated in Figure 7 and Appendix B. I would strongly suggest using an interpolation approach suited for highly managed systems (e.g. Eickenscheidt *et al.*, 2015). If this should not be possible due to inadequate PAR ranges during measuring campaigns, only campaign data (instead of annual balances) may be evaluated.

**Further comments**

**Line 59:** Better cite the most recent Dutch inventory data instead of an "old" (2009) paper.

**Table 1:**
- Details (e.g. SOC, clay content) on the "mineral top layer" would be helpful.
- Soil properties averaged for 0 to 70 cm are not really informative, better provide data on the topsoil and on depths where the water level/moisture changes actually occurred.

- How comparable are SSI and control when they partially strongly differ in SOM content (location D) or C:N ratio (location A)?
- Data on hydraulic conductivity or at least on the degree on decomposition are needed to discuss the contrasting hydrologic effects of SSI at the four locations. Does the "schalter" layer have any effect on the sites' hydrology?

**Line 140:** How was the C-export actually determined? For the frames (line 166 ff) or for the whole field as it is implied here? Were the frames fenced off from grazing? If the C-export had to be excluded from statistical analysis, how could the GHG balance containing the C-export be analysed?

**Line 145: Flux measurements and modelling**

While I understand that not all details can be provided to limit the length of the manuscript, lots of information is missing which would allow assessing the quality of the data.

- Were the chambers cooled and vented for pressure equilibration?
- Was PAR outside the chamber corrected for the light transmittance of the chamber before interpolating GPP? Which light transmittance was assumed/measured?
- Was there any quality control procedure for flux calculation (linearity, outliers, leakage…)?
- Was there any minimum temperature difference within one $R_{eco}$ campaign to avoid artefacts due to extrapolation of Eq. 2?
- Which $R_{eco}$ (nearest?) was subtracted from NEE to yield GPP (line 192)?
- The unit of $\alpha$ is wrong (line 200), it should be mg $CO_2$-C m$^{-2}$ h$^{-1}$ / μmol m$^{-2}$ s$^{-1}$
- Why did you choose to interpolate parameters and not weighted fluxes (line 204)?
- Why did you use $GPP_{max}$ and not $GPP_{opt}$ (Falge *et al.*, 2001) which is less susceptible to extrapolation errors?

**Line 254 ff: "Drainage" and "irrigation" periods**

From my understanding, "drainage" and "irrigation" periods are not defined correctly. While it remains unclear which WT (0.5, 1.5 or 3.0 m from the pipe) was used for this calculation, it is of course useful to differentiate whether SSI was dryer or wetter than the control when comparing $R_{eco}$. However, "drainage" and "irrigation" periods can only be identified by using absolute heads, i.e. by comparing the field WT to the ditch water level!

In this context, it also remains unclear why the SSI system works better at some of the fields in terms of hydrology – is there always enough water in the ditches, how is the hydraulic conductivity or at least the degree of decomposition of the peat, or are there strong differences in WT at 0.5, 1.5 and 3.0 m difference to the pipes which could be used to deduce information on the hydraulic conductivity?

Furthermore, it is rather difficult to compare results to other studies. Therefore, please give numbers for the mean and the summer mean water level.

**Table 2 and Table 3**

- Should be merged and N-fertilisation should be added.
- Uncertainties should be added.

**Line 425 ff:** There are some comparisons to other studies, but the authors do not try to explain the differences in emissions between their sites.

**Line 439:** How do you know that moisture conditions were optimal in 2017?

**Line 451:** What do you mean with "abnormal data points"?

**Line 486:** Effects of land-use intensity and land-use history should be discussed in the context of general emission level (section 4.3) as these aspects do not fit to the section "costs and benefits SSI".

Besides methodological issues, the manuscript seems to be hastily prepared which results in many inaccuracies especially regarding the references (list might not be exhaustive):

- Several references mentioned in the text are not in the list of references (Hoffmann *et al.*, 2015, Tiemeyer *et al.*, 2020)
- One reference appears twice (Berglund and Berglund, 2011)
- References are incomplete (Couwenberg, 2009, Tanneberger *et al.*, 2017)
- Generally, there is some tendency to cite non-peer reviewed literature (Joosten and Clarke, 2002, Joosten, 2009, Jurasinski *et al.*, 2016, Hendriks *et al.*, 2007b, Hoving *et al.*, 2015, van den Akker *et al.*, 2008, van den Born *et al.*, 2016). In many cases, peer-reviewed papers could easily be found and should be cited instead.

Furthermore, table and figure headings are often very brief or contain abbreviations, sometimes also such which are not used in the manuscript (e.g. location "*Ger"* in Appendix B).

**References**

Bader, C., Müller, M., Schulin, R., Leifeld, J., 2018. Peat decomposability in managed organic soils in relation to land-use, organic matter composition and temperature. Biogeosciences 15, 703–719. https://doi.org/10.5194/bg-15-703-2018

Eickenscheidt, T., Heinichen, J., Drösler, M., 2015. The greenhouse gas balance of a drained fen peatland is mainly controlled by land-use rather than soil organic carbon content. Biogeosciences 12, 5161-5184, https://doi.org/10.5194/bg-12-5161-2015

Flessa, H., Wild, U., Klemisch, M., Pfadenhauer, J.P., 1998. Nitrous oxide and methane fluxes from organic soils under agriculture. European Journal of Soil Science 49, 327-325, https://doi.org/10.1046/j.1365-2389.1998.00156.x

IPCC (Intergovernmental Panel on Climate Change), 2014. 2013 Supplement to the 2006 IPCC Guidelines for National Greenhouse Gas Inventories: Wetlands, Hiraishi, T., Krug, T., Tanabe, K., Srivastava, N., Baasansuren, J., Fukuda, M., Troxler, T.G. (eds), IPCC, Switzerland.

Koponen, H.T. and Martikainen, P.J., 2004. Soil water content and freezing temperature affect freeze–thaw related $N_2O$ production in organic soil. Nutrient Cycling in Agroecosystems 69, 213–219, https://doi.org/10.1023/B:FRES.0000035172.37839.24

Tiemeyer, B., Freibauer, F., Albiac Borraz, E., Augustin, J., Bechtold, M., Beetz, S., Beyer, C., Ebli, M., Eickenscheidt, T., Fiedler, S., Förster, C., Gensior, A., Giebels, M., Glatzel, S., Heinichen, J., Hoffmann, M., Höper, H., Jurasinski, G., Laggner, A., Leiber-Sauheitl, K., Peichl-Brak, M. & M. Drösler, 2020. A new methodology for organic soils in national greenhouse gas inventories: data synthesis, derivation and application. Ecological Indicators 105838, https://doi.org/10.1016/j.ecolind.2019.105838

---

## Author Comment (AC1) · 6 Sep 2020

Reviewer (R#1) comments and author responses to ms bg-2020-230

Reviewer comments are given in italic and with author responses in normal style

**Sub-soil irrigation does not lower greenhouse gas emission from drained peat meadows**

**by Stefan Weideveld et al.**

*General comments:*

*The authors investigate the GHG reduction potential of drained peatlands by using sub-soil irrigation. The topic of the paper is of relevance to Biogeosciences and will be of interest to an audience interested in mitigating greenhouse gas emissions from agriculturally used peatland. It is a novel approach, which needs further research. For the evaluation of the effect of sub soil irrigation on GHG emissions, a paired design of a control site and a sub-soil-irrigated site is used. Four different sites were investigated. $CO_2$, $CH_4$ and $N_2O$ fluxes were measured with chambers over a two 2 years period. Carbon and greenhouse gas budgets are determined and compared for the paired sites.*

> **Response(1):** We thank the reviewer for the positive comments and constructive inputs. This will help us improve the manuscript.

*I do not understand the experimental setup: The basic hypotheses of the manuscript that main GHG emissions comes from soil layers deeper than 70 cm is not well explained.*

> **Response(2):** In the Netherlands, the aim of the government is to reduce $CO_2$ emission from peat meadow areas by 1 Mt by 2030, from which halve is expected to be achieved with the SSI technique (PBL, 2018). To come to this reduction, an area of 50.000 ha with SSI drainage pipes are planned and a $CO_2$ reduction of 50% is expected from this area. This technique has, however, never been validated by measured $CO_2$ emission data. Expectations are based on pilots with only soil subsidence measurements. In these pilots a relation between lowest GWT and soil subsidence is found, therefore the elevation of summer GWT is expected to contribute most to the reduction of $CO_2$ emission. So, our hypothesis is based on the state of the SSI technique according to policy in practice.

*Moreover, no information about soil properties and soil moisture of this relevant depth are given in the manuscript. As these soil data are missing, it is not clear, which amount of soil organic carbon is exposed to oxygen due to the alterated ground-water level. Often the bulk density is low in deeper peat layers.*

> **Response(3):** We agree that the current table providing soil data is inadequate. In the revised version we will replace the averaged soil properties with data of a higher resolution per soil layer. More details will be provided on the mineral cover layer, the schalter layer, the degraded peat layer and the less degraded peat layer.

*It would be interesting to calculate the additional % of aerated carbon due to the alteration of the groundwater level and to compare it to GHG emissions.*

> **Response(4):** It would have been interesting to measure the change in soil moisture though out the and time with fluctuating water tables, in combination with soil oxygen. To see what the true effects are on the aeration of the soil as a result of the SSI. However, we did not measure it during this field experiment.

*Moreover, the authors should estimate if the small differences in groundwater level (<20 cm) can lead to a theoretical GHG reduction, which can be measured with this method (and associated uncertainties) and experimental set up.*

> **Response(5):** GWL are in summer even elevated for up to 60 cm difference. This is again not our own expectation, but the expectations are based on previous pilot studies which are now commonly accepted in policy.

*Unfortunately, soil moisture was only measured to soil depth of20cm. (At site A, C, D, soil moisture and temperature is measured only in the mineral soil cover). Moreover, these data are not presented in the paper, although the importance of soil moisture is discussed in the discussion section.*

> **Response(6):** Soil moisture data will be included in the table to expand the soil properties. This will be data from a sampling done during the Peak of the drought period indicating the effect of SSI throughout soil profile.

*The main conclusion that SSI does not lower GHG emissions cannot be drawn from the presented data as most of time the difference in ground water level between the treatments was relatively small. However, when the differences in ground water level were > 20 cm a reduction of GHG emissions was observed. In my view, the conclusion from this paper would be that a substantial increase in groundwater level is needed to allow large enough effects in the emissions to be measured.*

> **Response(7):** The current design of SSI, at a depth of -70 cm and spaced 6 or 5 meters apart, was not capable of raising the water table to a level to have a sufficient effect on the GHG emission. Even with a flexible ditch water level, inflow of water as not able to raise the water table to higher level. The conclusion was intended to state that the current way that the SSI was implemented does not allow for large enough effects on the groundwater table to have a measurable effect on the emission. Optimization of the SSI technique was not part of the main conclusion, but indeed a substantial higher water table is needed.

*For two sites, the comparability of control and SSI treatments is not given. This may influence the mineralization processes and thus the results. In particular, Site A: SSI has considerably higher organic matter content (39 vs 27%) as the control, and C/N ratios (29 vs 20) indicate different organic matter quality. Moreover, Site D: Control site has nearly the double amount of organic matter than SSI (38 vs 61%). This aspect is not discussed in the manuscript and might bias the results.*

> **Response(8):** The differences in organic matter content are largely due to the thickness of the mineral top layer. However, the soil organic carbon stock is of a similar size for both sites. To avoid confusion, the indication of soil organic matter will be changed into g/l soil. And it will be indicated for the different soil types to give a better sign of the comparability between the treatment and control and the different sites.

*In particular, the methods used to measure the carbon and greenhouse gas fluxes and management are not described in sufficient detail.*

> **Response(9):** The method is expanded upon, and described in further detail.

*The annual N2O budget was calculated based on only few measurement campaigns. In my opinion, it is not possible to calculate annual N2O budgets from 6-9 daily values, which were measured within 6 month in 2017. This is most evident at Site B Control: Linear interpolation of the high N2O emission in March probably overestimate the N2O emissions for the whole winter time. Moreover the material and methods section is misleadingly stating that N2O was measured for each measuring camping, but at Site B 38 campaigns were made, whereas I counted only 17 data points in Figure C1.Researchers reading the manuscript without looking at the supplementary data could extract the N2O data for annual budgets. Due to the low temporal resolution of the N2O data, it is not possible to distinguish between background N2O emissions and fertilizer-induced N2O emissions.*

**Response(10):** In 2017 we experienced infrastructural constraints to measure $N_2O$ fluxes more frequently. The extended winter gap is a consequence of mal-functioning of the Picarro 2508 under field conditions with low temperature. We agree that 7 flux days and 90 measurements are too few for year budget estimation. The methods and results will be adjusted so that it becomes clear that the year budget of 2017 is a rough estimation based on average fluxes from 7 flux days. We will discuss the importance of generally higher winter emissions that were not measured for year budget estimates, which made our estimation a conservative underestimation. However, we believe that the measured data is still valuable for evaluating the $N_2O$ emissions under influence of SSI. The results show no structural higher or lower $N_2O$ emissions between the control and SSI sites. The measured data fits our expectations and references of these types of systems. Therefore, we would like to keep the annual budget estimation for the general discussion of the total GWP. But clarification will be added to methodology and discussion the stress the low temporal resolution of our measurements, and daily measured data will be presented. The moments between frost and thaw was measured for Farm B and C in the beginning March 2018. However due to technical difficulties with low temperatures and the gas measure equipment these moments were still sparse.

*As no daily data are presented for the CH4 budget, the data coverage and thus quality of annual budget cannot be evaluated.*

**Response(11):** Daily data will be added to the manuscript to improve the data evaluation.

*The description of management is very short and important information about cutting days, fertilization events, and amount of applied fertilizer are missing, which makes it difficult to understand N2O and NEE data.*

**Response(12):** Cutting days and fertilization events will be added to Figure 7, Appendix B1, C1. Furthermore, fertilizer information will be included in the methods.

*E.g. Why are the cutting days not visible in the GPP data? In other studies, the decrease in GPP after management events can be nicely seen (Poyda et al. 2016 or Beetz et al 2013,). In comparison to their data, the GPP stayed rather constant and relatively high (-10 g CO2 m-2 d-1) throughout the year. Accordingly, it is difficult to evaluate the quality of GPP modeling.*

**Response (13):** We will re-calculate the GPP of 2017 campaign-wise with inclusion of the cutting events where the GPP will be reduced. This will result in a different GPP value and different figures 7 and appendix. Seeing the effect of the higher biomass on the dark respiration of the plants, we agree that the effect of the biomass is important for the total value of the GPP and Reco. In order to see the effect of cutting a correction for both

interpolations will be applied. The harvest dates will be included in the figures to visualize these moments. And to give a better estimate for the total emission.

*The uncertainty assessment is nicely done for the gap filling method of NEE, but the uncertainty estimates are not integrated in the results and transferred to GPP and Reco. For NEE, Reco, and GPP an uncertainty is indicated, but it is not stated what it is(error of SD or 95% confidence interval...). The uncertainty range is given for NEE as3-16 t CO2 ha-1 yr-1, (L 370), but NEE uncertainty from NEE gap filling is given as 14-25 t CO2 ha-1 yr-1). For N2O and CH4 the uncertainty assessment is missing. Other sources of uncertainty (systematic errors of the use of chamber methods or random errors) are not discussed. Please provide a more thorough uncertainty estimation of all component of the net ecosystem carbon balance and included this values in Table2 and*

> **Response(14):** Uncertainty will be discussed and quantified in more detail. Specifically, ranges of $R_{eco}$ and GPP will be presented by propagating model parameter variations to the gap-filled annual fluxes. Uncertainty from gap-filling method selection that was already discussed will be used as range of annual NEE, since we consider this as the major source of the uncertainty.

*3.Specific comments:*

*Experimental set up and management Table 1: please provide more information about soil properties of the mineral soil cover, and underlying peat layers (carbon content, bulk density, C/N and carbon stock) in a higher resolution for the entire aerated soil depth. Please state how many soil samples were taken per depth and where. Please also add the information of the depth location of the schalter.*

> **Response(15):** A table with higher resolution will be added to provide more information on the soil characteristics. The methodology will be updated.

*Figure 2: please provide information about the location of the chambers of the control site relative to the main ditch.*

> **Response(16):** The distance to the main ditch is added. The distance variated between 25 and 40 meters. The location was chosen to exclude a direct effect of the ditch on the water table in the control sites.

*Please provide data of cuttings days, fertilization events and measurement campaigns for CO2, N2O and CH4 as the growth of the grass and thus GPP strongly depend on time of measurement (days after cutting). Information can be added in Figure 7, Appendix B1, C1.*

> **Response(17):** We agree and in the revised version we will include the cutting days and fertilization events into the figures.

*Please add information about amount and determination of N und C input through slurry application. Please add information about the determination of the yield (dry mass of the grass).*

> **Response(18):** This information will be added in the methods and results. From every manure application manure samples were taken. Bulk-density was determined, Total nitrogen (TN) and total carbon (TC) was determined in dry slurry material (3 mg) using an elemental CNS analyzer (NA 1500, Carlo Erba; Thermo Fisher Scientific, Franklin, USA)

*According to Table 1 Site A und B were grazed. How was carbon import through cattle manure determined? How was the carbon export through grazing determined? Was the yield only determined within the chambers or for the entire grassland? How was the grass height determined?*

> **Response(19):** The management of the whole field was grazing, however our field site was fenced off to prevent the mentioned problems. The yield was determined inside the chamber frames, to close the carbon budget. Grass height was estimated using a straight scale with a plastic disk with a diameter of 30cm to determine the top of the grass.

*Gas fluxes Chambers: Was the location of the frames fixed over the two years? Did the vegetation change within the chambers during the experiment?*

> **Response(20):** The frames where fixed trough out the two measurement years. The vegetation though out the years remained dominated by *Lolium perenne.* However in spring there were always other species coming up in the frame. However after the first harvest these species disappeared.

*Please add the transparency of the chambers? Was a correction term introduced due to a reduced transparency?*

> **Response(21):** We corrected the PAR values outside the chamber since the acrylic glass of the transparent chambers reflected or absorbed at least 8% of the incoming radiation

*Please add information about the used sign convention, positive fluxes= loss of carbon? Please add information about the used equation for the calculation of GHG balances and assumptions (harvest is assumed to be released as CO2?, loss of dissolved organic carbon?*

> **Response(22):** The methodology is expanded upon. The atmospheric sign convention was used. All C fluxes into the ecosystem where defined as negative (uptake from the atmosphere into the ecosystem), and all C fluxes from the ecosystem to the atmosphere are defined as positive. This also holds for non-atmospheric inputs like manure (negative) and outputs like harvests (Positive). Both harvest and manure input are expected to be released as $CO_2$ again. This will be described better in the methods. Dissolved organic carbon was not sampled during the experiment.

*L242ff: what is the accuracy of precipitation data derived from satellite images?*

> **Response(23):** The accuracy is four square kilometer. Giving a precipitation value every 3 hours.

*L243: June 2017 seemed to have received more than the average precipitation June is included in the drought period?*

> **Response(24):** The average precipitation in June was higher than average, however this is due two days with heavy rain at the end of the month, ending the drought. We will include the dates of the determined drought periods.

*Figure 4: As there were 3 groundwater measurements per site, it is not clear which groundwater table is presented, average of all 3?, what is the SD of the three wells? How is the variability of groundwater level of the control site? Please explain DRN*

> **Response(25):** The presented data is data from the logger in the field site, the other groundwater measurements are manual dip wells, recorded each measurement campaign. The data shown in figure 4 is a good depiction of the situation in the control site. Only close

to the ditch (Less than 10 meters) there is a higher groundwater table in the summer and lower in the winter.

*L276: I do not understand the sentence "There is variation.." please clarify.*

**Response(26):** There is difference (variation) between the SSI and control site on the different days in regards to temperature and grass height.

*L327-330: What is meant by uncertainty of 3-16 t CO2 ha-1 yr-1. What is represented by 1.6 t CO2 ha-1 yr-1 I for NEE in 2017?L326-332: What is the difference between annual NEE of 47 t CO2 eq. ha-1 yr-1(L327) and emissions of 62 t CO2 eq. ha-1 yr-1 (L313)*

**Response(27)** This part will be rewritten to help clarify what parts are the uncertainty of the interpolation of NEE and what parts are the SD of the NEE. What parts are the NEE budget (GPP – $R_{eco}$) and what part are the total carbon budget.

*L 334-338: Please provide daily CH4 data.*

**Response(28):** Daily data will be added to the manuscript

*Table 2 and Table 3:  Please add uncertainty estimates for all components of GHG balance.*

**Response(29):** Modeling and gap-filling uncertainties will be added to $R_{eco}$, GPP and NEE.

*L. 380: Reco was lower when the differences of groundwater level was >20 cm*

**Response(30):** Correct, this will be adjusted in the manuscript.

*L. 420: N2O emissions are not only driven by fertilization events, but also by soil moisture, which should be differ by the treatment.  Thus, the comparison can be biased by missing peak events.*

**Repsponse(31):**See response(10) Soil moisture is an important driver for the N2O fluxes from these drained peatland systems. We assume that with the method used we missed peaks induced by fertilization and rewetting. However a comparison between the treatments effects on the basis of the different measurement campaigns can still provide insight into the effect of SSI on $N_2O$ emissions.

*L428: please use the same sign convention for all cited references.*

**Response(32):**  The references will be updated and check more thoroughly.

*Technical comments: L310: Please state was the 4 t are, SD?,...,*

**Response(33):** this will be clarified in the manuscript. In this case it is the SD

*Please indicate A und B in Figure 6*

**Response(33):** A and B will be included in figure 6

*Figure 7: please use colors, which can be clearly distinguished*

**Response(34):** Figure 7 will be improved to increase the understandability of the figure.

---

## Author Comment (AC2) · 6 Sep 2020

Reviewer (R#2) comments and author responses to ms bg-2020-230

Reviewer comments are given in italic and with author responses in normal style

**Sub-soil irrigation does not lower greenhouse gas emission from drained peat meadows**

**by Stefan Weideveld et al.**

*Generally, the manuscript will be of interest for readers of Biogeosciences, and the topic of adequate mitigation strategies for drained organic soils is one of high relevance. While the overall result that there is no difference in GHG emissions of this sub-surface irrigation (SSI)system and the control seems to be robust, there are, in my opinion, still four major issues which need to be solved before the manuscript could be considered for publication in BG:*

> **Response (1)** We thank the reviewer for the positive comments and constructive inputs. This will help us improve the manuscript.

• *The authors appear to be surprised that SSI does not result in lowered GHG emissions, but this "surprise" is rather unfounded as the water table is raised only slightly towards a target level of 60 cm below ground, which I would –in line with the IPCC Wetlands Supplement (IPCC, 2014) –still regard as "deeply drained".*

> **Response (2)** We recognize that part of the questions raised are a result of an inadequate framing of the experiment. This is an important factor for the paper, to improve this the current state of the SSI technique will be given. In the Netherlands, the aim of the government is to reduce $CO_2$ emission from peat meadow areas by 1 M t by 2030, from which halve is expected to be achieved with the SSI technique (PBL, 2018). To come to this reduction, an area of 50.000 ha with SSI drainage pipes are planned and a $CO_2$ reduction of 50% is expected from this area. However, the current design of the SSI technique aims to increase the lowest water table while maintaining the agricultural function as "business as usual". Also, this technique has never been validated by measured $CO_2$ emission data. Expectations are based on pilots with only soil subsidence measurements. In these pilots a relation between lowest GWT and soil subsidence is found, therefore the elevation of summer GWT is expected to contribute most to the reduction of $CO_2$ emission. The current set-up tested in our experiment aims to explore the effectiveness of SSI on GHG emission for the first time by measurements, also on a large scale on sites representative for the Frisian peat meadows. Therefore, our hypothesis is based on the state of the SSI technique according to policy in practice. And our set-up was made based on the current policy status rather than the scientific exploration of the optimal use of rewetting to mitigate the emissions.

• *According to Figure C1, there were partially only 7 measurement dates for N2O in 2017 and afterwards a gap of five months. Given the highly episodic nature of N2O fluxes, this is absolutely inadequate for the calculation of annual balances in a strongly fertilized grassland.*

> **Response (3):** In 2017 we experienced infrastructural constraints to measure $N_2O$ fluxes more frequently. The extended winter gap is a consequence of mal-functioning of the Picarro 2508 under field conditions with low temperature. We agree that 7 flux days and 90 measurements are too few for year budget estimation. The methods and results will be adjusted so that it becomes clear that the year budget of 2017 is a rough estimation based on average fluxes from 7 flux days. We will discuss the importance of generally higher winter emissions that were not measured for year budget estimates, which made our estimation a conservative underestimation. However, we believe that the measured data is still valuable for evaluating

the $N_2O$ emissions under influence of SSI. The results show no structural higher or lower $N_2O$ emissions between the control and SSI sites. The measured data fits our expectations and references of these types of systems. Therefore, we would like to keep the annual budget estimation for the general discussion of the total GWP. But clarification will be added to methodology and discussion the stress the low temporal resolution of our measurements, and daily measured data will be presented. The moments between frost and thaw was measured for Farm B and C in the beginning March 2018. However due to technical difficulties with low temperatures and the gas measure equipment these moments were still sparse.

- *For the interpolation of GPP, all measurement campaigns have been pooled for 2017 and harvests have not been accounted for when interpolating GPP despite the large influence of above ground biomass on maximum photosynthetic rates.*

> **Response (4):** We appreciate the reviewer's suggestions for improved gap-filling strategies. The data for GPP gap-filling is available for a recalculation of the GPP balance for 2017 for a better estimate of the GPP. Pooling of measurement campaigns will be improved based on conditions during the measurements. The amount of biomass and the harvest are key in understanding the GPP flux. We will include the harvest moments in the interpolations of GPP, and consider possible need of correction for the Interpolated Reco regarding effect of the amount of biomass on dark respiration.

**Title and assumption that this specific SSI system would lower GHG emissions**

*The SSI system studied here has a target water level of -60 cm. Given the limited hydraulic conductivity of the peat and the "exit resistance" of the pipes, a water level of -60 cm in the ditches results in even deeper field water levels in summer. This target seems to be based on the assumption that CO2emissions originate from deeper peat (see below). Thus, the authors state that a WT rise of 6-18 cm in summer compared to an even lower level "unexpectedly" (line 22) or "contrary to our expectations" (line 29) does not lower GHG emissions. In my opinion, this is absolutely no surprise, but should be expected as laboratory studies often show highest respiration rates at medium water content and as field studies, on average, showed an asymptotic rather than a linear response of CO2 emissions to water table depth (too dry, no more peat exposed, Tiemeyer et al., 2020).*

> **Response (5)**: See response(2) for a full response. This is not our own expectation, but the expectations are based on previous pilot studies and now common accepted in policy.

*Thus, the title needs to be changed to "Sub-soil irrigation with target water levels of 60 cm does not lower carbon dioxide emissions from drained peat meadows" or something similar, as the experiments do not allow for conclusions on SSI in general. Further, if the authors are really surprised by their results, they will need to convince the reader why. In this context, it also needs to be discussed why such low target water levels have been chosen at all. At least for meadow use as in 2018, such low water levels are technically not needed when adequate machinery (low weight, double tyres, etc.) is used.*

> **Response (6):** A change will be made in the title and conclusion, to clarify that the current design of SSI is the commonly applied compromise between additional drainage and increased infiltration during summer and that this technique may fall short to have a significant effect on the GHG balance. Furthermore, information will be added in regard to the average ditch water level to indicate that the goal of a water table of -60 was further promoted by raising the ditchwater in de summer periods, on average the ditch water level connected to the SSI was closer to -40 cm rather than -60 cm.

***Peat layers below -70 cm contribute most to GHG emissions***

*In the introduction, there is no reasoning why this should be the case at all. Many studies have shown that top soils show higher respiration rates than subsoils e.g. due to higher nutrient contents or generally more favourable conditions for microbial activity(e.g. Bader et al., 2018). This is indeed briefly discussed on page 22, but the whole "story"of the manuscript (and probably also the design of the sub-surface irrigation system) builds on this assumption. Thus, either it needs to be substantiated by peer-reviewed (!) literature, or the manuscript needs to be restructured based on more adequate hypotheses.*

> **Response (7):** We agree with the reviewer that the manuscript needs a clear distinction between current knowledge in peatland sciences and (current) assumptions of land authorities and Dutch governmental institutions responsible for emissions reporting from peatlands. In our own scientific reporting (van den Berg et al. 2018) we show that the top 20 cm of peat revealed the highest $CO_2$ production potential. In contrast, the current estimation methods in the Netherlands make no use of $CO_2$ flux data but rely on soil volume – soil carbon models. It is assumed that soil subsidence is quasi 1:1 related to carbon losses in form of $CO_2$ without taking volume changes of the peat and changes in the carbon density into account. Based on that 1:1 soil subsidence-soil carbon relationship it has been inferred that soil subsidence is stronger when groundwater resides during summer

**Frequency of N2O flux measurements**

*According to FigureC1, there seem to be only 7 measurement dates for N2O in some cases in 2017, then a gap of more than 5 months in winter and finally a further gap of two months at the end of the study period. This contradicts the text that N2O was measured at each campaign, i.e. supposedly bi-weekly in summer and monthly in winter(page 8). If Figure C1 is actually correct, this data may not be used for the calculation of annual balances as effects of fertilisation cannot be captured adequately with such a low temporal resolution. Further, I would suspect that the first fertilisation event took place before April and was thus missed by the campaigns. In any case, fertilisation dates should be indicated in Figure C1.*

*Even more important, it is well-known that high N2O emissions may occur when temperatures change between frost and thaw(e.g. Koponen and Martikainen, 2004), especially under wetter conditions, and that maximum N2O fluxes of drained peatlands may occur in winter also under temperate climatic conditions (e.g. Flessa et al., 1988). Therefore, the authors should refrain from calculating annual balances from a dataset without winter data. The N2O data could, however, be used to compare treatment effects on the basis of campaigns. In consequence, this means that GHG balances cannot be calculated from the presented data, but only C balances.*

> **Response (8):** See response (3) for the elaboration about the $N_2O$ choices that were made in the manuscript and the changes that we will make.

*GPP modelling*

*In my opinion, pooling all summer data as done for 2017 is not an adequate gap-filling strategy as GPP max and $\alpha$ strongly depend on vegetation development. This strategy of pooling might be valid for (semi-)natural vegetation, but no for intensively used grasslands with frequent harvests.*

*Further, it seems that parameters are generally interpolated across harvests which does not capture the effects on GPP, which should be very low after harvests. Harvests are unfortunately not indicated in Figure 7 and Appendix B. I would strongly suggest using an interpolation approach suited for highly managed systems (e.g. Eickenscheidt et al., 2015). If this should not be possible due to inadequate PAR ranges during measuring campaigns, only campaign data (instead of annual balances) may be evaluated.*

> **Response (9):** We will re-calculate the GPP of 2017 campaign-wise. We will also include the cutting events as reducing the GPP. This will result in a different GPP value and different figures 7 and appendix. We agree that the effect of the biomass is important for the total value of the GPP and Reco. In order to see the effect of cutting a correction for both interpolations is needed. Seeing the effect of the higher biomass on the dark respiration of the plants. The harvest dates will be included in the figures to visualize these moments. And to give a better estimate for the total emission.

**Further comments**

*• Line 59: Better cite the most recent Dutch inventory data instead of an "old" (2009) paper.*

> **Response(10):** The most recent Dutch inventory will be used to have an indication of the national emissions from drained peatlands.

*Table 1:*

*• Details (e.g. SOC, clay content) on the "mineral top layer" would be helpful.• Soil properties averaged for 0 to 70 cm are not really informative, better provide data on the top soil and on depths where the water level/moisture changes actually occurred.*

> **Response(11):** We agree that the current table is inadequate. In the revised version we will replace the averaged soil properties for a higher resolution per soil layer. More details will be provided on the mineral cover layer, the schalter layer, the degraded peat layer and the less degraded peat layer. The mineral content was determined, however the fractions of the mineral top layer where not determined.

*• How comparable are SSI and control when they partially strongly differ in SOM content (location D) or C:N ratio (location A)?*

> **Response(12):** The differences in organic matter is largely due to the thickness of the mineral top layer. However, for the soil organic carbon stock is of a similar size for both sites. The soil organic matter will be indicated as g/l soil.

*• Data on hydraulic conductivity or at least on the degree on decomposition are needed to discuss the contrasting hydrologic effects of SSI at the four locations.*

> **Response**(13): The hydraulic conductivity was not measured during the experiment. However, the dip wells that we used to measure the water table for the different frames could give an indication for the processes in the field. Only one location had a good horizontal water flow in the peat layer, that was location B. Location A saw a strong effect of the SSI in the water table. However, in some places there was a large difference in the water table at different distances from the pipes.

• *Does the "schalter" layer have any effect on the sites' hydrology?*

> **Response(14):** Schalter is known to limit vertical water flow, due to its laminated structure. However, there is little documented about the properties and processes. In our case, the locations with "schalter" seem to have lower effects from the SSI

• *Line 140: How was the C-export actually determined? For the frames (line 166 ff) or for the whole field as it is implied here? Were the frames fenced off from grazing?*

> **Response(15):** The experiment sites were fenced off from grazing. The C-Export was determined inside the frames. However, on harvest days the whole experimental site was cut.

• *If the C-export had to be excluded from statistical analysis, how could the GHG balance containing the C-export be analysed?*

> **Response(16):** The C-Export has been set for a similar amount for both fields. The NEE, $CH_4$ and $N_2O$ budgets will be focused in the statistical analysis instead of the total GHG balance.

*Line 145: Flux measurements and modelling*

*While I understand that not all details can be provided to limit the length of the manuscript, lots of information is missing which would allow assessing the quality of the data.*

• *Were the chambers cooled and vented for pressure equilibration?*

> **Response(17):** The chambers where not cooled. The pressure inside was equilibrated when placing the chamber on the frames.

• *Was PAR outside the chamber corrected for the light transmittance of the chamber before interpolating GPP? Which light transmittance was assumed/measured?*

> **Response(18):** We corrected the PAR values outside the chamber since the acrylic glass of the transparent chambers reflected or absorbed at least 8% of the incoming radiation

• *Was there any quality control procedure for flux calculation (linearity, outliers, leakage...)?*

> **Response(19)**: Each flux was checked, for the dark measurements a only fluxes with a R-squared of 0.99 or higher where used. For the light measurements the majority of the fluxes that were used had a R-squared of 0.95. The exception where the fluxes with slopes close to zero or zero (equilibrium between gross primary production – GPP – and $R_{ECO}$) were not discarded.

• *Was there any minimum temperature difference within one Reco campaign to avoid artefacts due to extrapolation of Eq. 2?*

> **Response(20):** There was no minimum temperature difference set within the Reco campaign. The measurement campaigns where planned to have a range in light variation for the GPP calculations, this resulted in a good temperature range during the day.

• *Which Reco(nearest?) was subtracted from NEE to yield GPP (line 192)?*

> **Response(21):** The Reco closest in time was used for subtraction of NEE to yield GPP. During the campaigns light and dark measurements were always conducted in the same time frame.

• *The unit of $\alpha$ is wrong (line 200), it should be mg CO2-C m-2h-1/ μmol m-2s-1*

**Response(22):** Is adjusted in the material and methods.

• *Why did you choose to interpolate parameters and not weighted fluxes(line 204)?*

**Response**(23): We interpolated the parameters by assuming a linear development of the parameters between two measurement campaigns. E.g. a linear development of the temperature sensitivity of $R_{eco}$.

• *Why did you use GPPmax and not GPPopt(Falge et al., 2001)which is less susceptible to extrapolation errors?*

**Response (24):** According to Falge et al. (2001), GPPmax (or saturation value of GPP) has less explanatory worth for real systems since PAR will not reach infinite, therefore the author switched to GPPopt which provides a reference value at a certain PAR level. GPPmax was used in numerous fluxes modeling works, and we did not find argument from literature stating significantly larger uncertainty from the use of GPPmax. Thereby, if a GPPopt is used, there should have enough data in a specific PAR value (e.g. 2000 $\mu mol\ m^{-2}\ s^{-1}$). With eddy covariance (where this gap filling paper of Falge et al. is written for) this is not a problem, but with chamber measurement data is limited. The accuracy will therefore be better to fit the light response curve with the GPPmax.

**Line 254 ff: "Drainage" and "irrigation" periods**

• *From my understanding, "drainage" and "irrigation" periods are not defined correctly. While it remains unclear which WT (0.5, 1.5 or 3.0 m from the pipe) was used for this calculation, it is of course useful to differentiate whether SSI was dryer or wetter than the control when comparing Reco. However, "drainage" and "irrigation" periods can only be identified by using absolute heads, i.e. by comparing the field WT to the ditch water level!*

**Response(25):** We will clarify the definition of these periods. 'Drainage' periods refer to moments when there is drainage to the ditch and 'irrigation' periods refer to moments with water infiltration from the ditch to the field. Here the aim is to differentiate between SSI and control.

• *In this context, it also remains unclear why the SSI system works better at some of the fields in terms of hydrology –is there always enough water in the ditches, how is the hydraulic conductivity or at least the degree of decomposition of the peat, or are there strong differences in WT at 0.5, 1.5 and 3.0 m difference to the pipes which could be used to deduce information on the hydraulic conductivity?*

**Response(26):** The water table in the ditch was maintained at a level between -60 to -20 cm from the soil surface. It was never a limiting factor for the functioning of SSI. The hydraulic conductivity of the peat soil was not measured during the experiment. However the functioning of the SSI gives an indication of the conductivity. This is closely related to the type of peat present. Farm A, C and D all have Sphagnum peat, with the layer where the pipe is present being moderately decomposed (H5-H7). We suspect there are some macro cracks in the peat soil of farm A, that help infiltration. For location B the peat soil consists of Alder peat. The layer where the pipe is present is moderately decomposed but with a large presence of wood/branches. For this location the SSI seems to work best. With a strong drainage and infiltration effect.

*• Furthermore, it is rather difficult to compare results to other studies. Therefore, please give numbers for the mean and the summer mean water level.*

**Response(27):** The mean annual average GWT table will be given to increase the comparability between the different sites and to other studies.

***Table 2 and Table 3***

*•Should be merged and N-fertilisation should be added. •Uncertainties should be added.*

**Response(28):** The tables will be merged and N-fertilisation is added to table 2/3. Modeling and gap-filling uncertainties will be added to $R_{eco}$, GPP and NEE.

*• **Line 425 ff:** There are some comparisons to other studies, but the authors do not try to explain the differences in emissions between their sites.*

**Response(29):** The comparison between the sites is expanded upon. The differences between the sites are largely because of the soil conditions. The locations with a mineral topsoil seem to respond stronger to drought. Furthermore, there was a difference between the starting conditions of the sites. The sites A and B where grazed before the experiment and site C and D where only mown. This resulted in a difference in the grass structure, where the grazed grass forms a more dense vegetation structure than the mown grass.

*• **Line 439 :** How do you know that moisture conditions were optimal in 2017?*

**Response(30):** The indication of 'optimal' come from observation of the conditions in the field, for example the grass growth that we observed during the field experiment. This was also determined in contact with the farmers who judged a better year for grass growth. We will reconsider the wording 'optimal', but the point was that we expect that the moisture levels were not a limiting factor during this summer period.

*• **Line 451:** What do you mean with "abnormal data points"?*

**Response(31):** The "abnormal data points" refer to measurement that did not fit into the temperature dependent function of $R_{eco}$ or light response curve of GPP, due to the extreme drought that limited soil respiration's response to higher temperature, or reduced the photosynthetic rate.

*• **Line 486** : Effects of land-use intensity and land-use history should be discussed in the context of general emission level (section 4.3) as these aspects do not fit to the section "costs and benefits SSI".*

**Response (32):** We chose to discuss about land use in the 'cost and benefit' section because of the possibility of SSI to be beneficial for the intensive land use. Due to the increased load boarding capacity of the fields and the drainage in Spring and Autumn, it is possible to extend the periods that the field can be managed. We consider this as a possible benefit from the SSI, however we didn't observe this during the experiment.

*• Besides methodological issues, the manuscript seems to be hastily prepared which results in many inaccuracies especially regarding the references (list might not be exhaustive): • Several references mentioned in the text are not in the list of references (Hoffmann et al., 2015, Tiemeyer et al., 2020)• One reference appears twice (Berglund and Berglund, 2011)•References are incomplete (Couwenberg, 2009, Tanneberger et al., 2017)*

**Response(33):** The references will be updated and check more thoroughly.

*• Generally, there is some tendency to cite non-peer reviewed literature (Joosten and Clarke, 2002, Joosten, 2009, Jurasinski et al., 2016, Hendriks et al., 2007b, Hoving et al., 2015, van den Akker et al., 2008, van den Born et al., 2016). In many cases, peer-reviewed papers could easily be found and should be cited instead.*

**Response(34) :** The current references will be updated. The choice for the non-peer reviewed literature is largely due to the current condition that many of the decisions made for the SSI-experiment by the local government were based on these references. And some indicate the aim of the national and provincial government to implement SSI on a large scale as a way of mitigating problems that occur with management of these Peat meadows.

*• Furthermore, table and figure headings are often very brief or contain abbreviations, sometimes also such which are not used in the manuscript(e.g. location "Ger" in Appendix B).*

**Response(35):** The table and figure headings are expanded upon the improve the understandability of the figures and the abbreviations will be written full out in the headings.

*References*

*Bader, C., Müller, M., Schulin, R., Leifeld, J., 2018. Peat decomposability in managedorganic soils in relation to land-use, organic matter composition and temperature.Biogeosciences 15, 703–719. https://doi.org/10.5194/bg-15-703-2018Eickenscheidt, T., Heinichen, J., Drösler, M., 2015. The greenhouse gas balance of a drained fen peatland is mainly controlled by land-use rather than soil organic carbon content. Biogeosciences 12, 5161-5184, https://doi.org/10.5194/bg-12-5161-2015Flessa, H., Wild, U., Klemisch, M., Pfadenhauer, J.P., 1998. Nitrous oxide and methane fluxes from organic soils under agriculture. European Journal of Soil Science 49, 327-325, https://doi.org/10.1046/j.1365-2389.1998.00156.xIPCC (Intergovernmental Panel on Climate Change), 2014. 2013 Supplement to the 2006 IPCC Guidelines for National Greenhouse Gas Inventories: Wetlands, Hiraishi, T., Krug,T., Tanabe, K., Srivastava, N., Baasansuren, J., Fukuda, M., Troxler, T.G. (eds), IPCC, Switzerland.Koponen, H.T. and Martikainen, P.J., 2004. Soil water content and freezing temperature affect freeze–thaw relatedN2O production in organic soil. Nutrient Cycling in Agroecosystems 69,213–219, https://doi.org/10.1023/B:FRES.0000035172.37839.24Tiemeyer, B., Freibauer, F., Albiac Borraz, E., Augustin, J., Bechtold, M., Beetz, S., Beyer, C., Ebli, M., Eickenscheidt, T., Fiedler, S., Förster, C., Gensior, A., Giebels, M., Glatzel, S., Heinichen, J., Hoffmann, M., Höper, H., Jurasinski, G., Laggner, A., Leiber-Sauheitl, K., Peichl-Brak, M. & M. Drösler, 2020. A new methodology for organic soils in national greenhouse gas inventories: data synthesis, derivation and application. Ecological Indicators 105838, https://doi.org/10.1016/j.ecolind.2019.105838*

---

## Author Comment (AC3) · 6 Sep 2020

*Having been involved in the latest discussions on the research presented in this paper, I would like to post 2 comments:*

> **Response(1)** We thank the Jos Schouwenaars for the positive comments and constructive inputs. This will help us improve the manuscript.

*1. In 2019, on some of these study sites, additional emission measurements have been made, using eddy covariance techniques. These measurements indicate far lower emissions as the ones with closed chambers, used in this study. It was recognized by the researchers that this might be due to an erroneous gap filling procedure. This should be mentioned and explained in this paper.*

> **Response(2)** The uncertainty of the interpolation will be updated, to give a better estimate of the restriction of the used method and gap filled procedures. However one of the eight sites was measured using Eddy covariance technique in a different year than presented in this manuscript.

*2. The results do not indicate a difference in emissions rates between the sites with subsurface infiltration and the control sites. It should be mentioned that this conclusions is valid for the design of the infiltration and the soil type as used in these experiments, i.e. - drains at a depth of 60 cm and at a distance of 6 meter. - sphagnum peat type with a very low permeability. These additional remarks are required because, despite the obvious results for these study sites, it might be possible that subsurface drainage leads to lower emission rates when applied with alternative designs ( e.g. lower depth, smaller distances, in soils with higher permeability, with addional water reservoirs to increase water pressure etc. ). Consequently it also is important to adapt the title of the paper.*

> **Response(3)** A change will be made in the title and conclusion, to clarify that the current design of SSI is the commonly applied compromise between additional drainage and increased infiltration during summer and that this technique may fall short to have a significant effect on the GHG balance. The design being at a depth of -70 and spaced 6 or 5 meters apart. Furthermore, information will be added in regard to the average ditch water level to indicate that the goal of a water table of -60 was further promoted by raising the ditchwater in de summer periods, on average the ditch water level connected to the SSI was closer to -40 than -60.
>
> We agree that the functioning of the SSI is closely related to the type of peat present. Farm A, C and D all have Sphagnum peat, with the layer where the pipe is present being moderately decomposed (H5-H7). We suspect there are some macro cracks in the peat soil of farm A, that help infiltration.  For location B the peat soil consists of Alder peat. The layer where the pipe is present is moderately decomposed but with a large presence of wood/branches. For this location the SSI seems to work best. With a strong drainage and infiltration effect.
>
> Further research is needed to see if other SSI designs can be adjusted to have a significant effect on the emissions.

---

## Author Response (AR1)

**Author's response**

Reviewer (R#1) comments and author responses and changes to ms bg-2020-230

We highly appreciate the very helpful and constructive comments of the anonymous referee, which helped us to further improve the manuscript. We tried to consider all of them.

Reviewer comments are given in italic and with author responses in normal style

**Sub-soil irrigation does not lower greenhouse gas emission from drained peat meadows**

**by Stefan Weideveld et al.**

*General comments:*

*The authors investigate the GHG reduction potential of drained peatlands by using sub-soil irrigation. The topic of the paper is of relevance to Biogeosciences and will be of interest to an audience interested in mitigating greenhouse gas emissions from agriculturally used peatland. It is a novel approach, which needs further research. For the evaluation of the effect of sub soil irrigation on GHG emissions, a paired design of a control site and a sub-soil-irrigated site is used. Four different sites were investigated.CO2, CH4 and N2O fluxes were measured with chambers over a two 2 years period. Carbon and greenhouse gas budgets are determined and compared for the paired sites.*

> **Response(1):** We thank the reviewer for the positive comments and constructive inputs. This helped us improve the manuscript.

*I do not understand the experimental setup: The basic hypotheses of the manuscript that main GHG emissions comes from soil layers deeper than 70 cm is not well explained.*

> **Response(2):** In the Netherlands, the aim of the government is to reduce $CO_2$ emission from peat meadow areas by 1 Mt by 2030, from which halve is expected to be achieved with the SSI technique (PBL, 2018). To come to this reduction, an area of 50.000 ha with SSI drainage pipes are planned and a $CO_2$ reduction of 50% is expected from this area. This technique has, however, never been validated by measured $CO_2$ emission data. Expectations are based on pilots with only soil subsidence measurements. In these pilots a relation between lowest GWT and soil subsidence is found, therefore the elevation of summer GWT is expected to contribute most to the reduction of $CO_2$ emission. So, our hypothesis is based on the state of the SSI technique according to policy in practice. The current state of application and the basic hypothesis are explained in introduction (L71 − 89).

*Moreover, no information about soil properties and soil moisture of this relevant depth are given in the manuscript. As these soil data are missing, it is not clear, which amount of soil organic carbon is exposed to oxygen due to the alterated ground-water level. Often the bulk density is low in deeper peat layers.*

> **Response(3):** We agree that the table providing soil data was inadequate. We replaced the averaged soil properties with data of a higher resolution per soil layer. More details are provided on the mineral

cover layer, the schalter layer, the degraded peat layer and the less degraded peat layer in Appendix B. (Page 31)

*It would be interesting to calculate the additional % of aerated carbon due to the alteration of the groundwater level and to compare it to GHG emissions.*

> **Response(4):** It would have been interesting to measure the change in soil moisture though out the and time with fluctuating water tables, in combination with soil oxygen. To see what the true effects are on the aeration of the soil as a result of the SSI. However, we did not measure it during this field experiment.

*Moreover, the authors should estimate if the small differences in groundwater level (<20 cm) can lead to a theoretical GHG reduction, which can be measured with this method (and associated uncertainties) and experimental set up.*

> **Response(5):** GWL are in summer even elevated for up to 60 cm difference. This is again not our own expectation, but the expectations are based on previous pilot studies which are now commonly accepted in policy. The basis of this expectation is now explained in detail (L71 – 89)

*Unfortunately, soil moisture was only measured to soil depth of20cm. (At site A, C, D, soil moisture and temperature is measured only in the mineral soil cover). Moreover, these data are not presented in the paper, although the importance of soil moisture is discussed in the discussion section.*

> **Response(6):** Soil moisture data is now included in the table in appendix B (Page 31) to expand the soil properties. This is data from a sampling done during the peak of summer period indicating the effect of SSI throughout the soil profile.

*The main conclusion that SSI does not lower GHG emissions cannot be drawn from the presented data as most of time the difference in ground water level between the treatments was relatively small. However, when the differences in ground water level were > 20 cm a reduction of GHG emissions was observed. In my view, the conclusion from this paper would be that a substantial increase in groundwater level is needed to allow large enough effects in the emissions to be measured.*

> **Response(7):** The current design of SSI, at a depth of -70 cm and spaced 6 or 5 meters apart, was not capable of raising the water table to a level to have a sufficient effect on the GHG emission. Even with a flexible ditch water level, inflow of water as not able to raise the water table to higher level (L509). The conclusion was intended to state that the current way that the SSI was implemented does not allow for large enough effects on the groundwater table to have a measurable effect on the emission. Optimization of the SSI technique was not part of the main conclusion, but indeed a substantial higher water table is needed (L513).

*For two sites, the comparability of control and SSI treatments is not given. This may influence the mineralization processes and thus the results. In particular, Site A: SSI has considerably higher organic matter content (39 vs 27%) as the control, and C/N ratios (29 vs 20) indicate different organic matter quality. Moreover, Site D: Control*

*site has nearly the double amount of organic matter than SSI (38 vs 61%). This aspect is not discussed in the manuscript and might bias the results.*

**Response(8):** The differences in organic matter content are largely due to the thickness of the mineral top layer. However, the soil organic carbon stock is of a similar size for both sites. To avoid confusion, the indication of soil organic matter is changed into g/l soil (Table 1, Page 5). And it is now indicated for all the different soil types to give a better sign of the comparability between the treatment and control and the different sites. Appendix B (Page 31)

*In particular, the methods used to measure the carbon and greenhouse gas fluxes and management are not described in sufficient detail.*

**Response(9):** The method is expanded upon, and described in further detail (L166).

*The annual N2O budget was calculated based on only few measurement campaigns. In my opinion, it is not possible to calculate annual N2O budgets from 6-9 daily values, which were measured within 6 month in 2017. This is most evident at Site B Control: Linear interpolation of the high N2O emission in March probably overestimate the N2O emissions for the whole winter time. Moreover the material and methods section is misleadingly stating that N2O was measured for each measuring camping, but at Site B 38 campaigns were made, whereas I counted only 17 data points in Figure C1.Researchers reading the manuscript without looking at the supplementary data could extract the N2O data for annual budgets. Due to the low temporal resolution of the N2O data, it is not possible to distinguish between background N2O emissions and fertilizer-induced N2O emissions.*

**Response(10):** In 2017 we experienced infrastructural constraints to measure $N_2O$ fluxes more frequently. The extended winter gap is a consequence of mal-functioning of the Picarro 2508 under field conditions with low temperature. We agree that 7 flux days and 90 measurements are too few for year budget estimation. We present an average measured N2O flux in table 4 (Page 21). The methods and results will be adjusted so that it becomes clear that measurements of 2017 are a rough estimation based on average fluxes from 7 flux days (L78). However, we believe that the measured data is still valuable for evaluating the $N_2O$ emissions under influence of SSI. The results show no structural higher or lower $N_2O$ emissions between the control and SSI sites. The measured data fits our expectations and references of these types of systems. Clarification be added to methodology and discussion (L494) to stress the low temporal resolution of our measurements, and daily measured data will be presented. The moments between frost and thaw was measured for Farm B and C in the beginning March 2018. However due to technical difficulties with low temperatures and the gas measure equipment these moments were still sparse.

*As no daily data are presented for the CH4 budget, the data coverage and thus quality of annual budget cannot be evaluated.*

**Response(11):** Daily CH4 data is added to the appendix D (Page 35) of the manuscript to improve the data evaluation.

*The description of management is very short and important information about cutting days, fertilization events, and amount of applied fertilizer are missing, which makes it difficult to understand N2O and NEE data.*

> **Response(12):** Cutting days and fertilization events are added to Figure 7, Appendix B1, C1. Furthermore, fertilizer information is included in the methods. (L229)

*E.g. Why are the cutting days not visible in the GPP data? In other studies, the decrease in GPP after management events can be nicely seen (Poyda et al. 2016 or Beetz et al 2013,). In comparison to their data, the GPP stayed rather constant and relatively high (-10 g CO2 m-2 d-1) throughout the year. Accordingly, it is difficult to evaluate the quality of GPP modeling.*

> **Response (13):** We re-calculated the GPP for the 2017 and 2018 campaign-wise 1) with data from adjacent campaigns clustered; 2) with inclusion of the cutting events where the model parameters ($R_{eco,Tref}$, $GPP_{max}$, and α) are reduced based on linear relationships between grass height and model parameters. In this way, better model performance is achieved and the influence from plant biomass is accounted for (L229). The harvest dates are included in figure 7 and in appendix C to visualize these moments. And to give a better estimate for the total emission.

*The uncertainty assessment is nicely done for the gap filling method of NEE, but the uncertainty estimates are not integrated in the results and transferred to GPP and Reco. For NEE, Reco, and GPP an uncertainty is indicated, but it is not stated what it is(error of SD or 95% confidence interval...). The uncertainty range is given for NEE as3-16 t CO2 ha-1 yr-1, (L 370), but NEE uncertainty from NEE gap filling is given as 14-25 t CO2 ha-1 yr-1). For N2O and CH4 the uncertainty assessment is missing. Other sources of uncertainty (systematic errors of the use of chamber methods or random errors) are not discussed. Please provide a more thorough uncertainty estimation* of all component of the net ecosystem carbon balance and included this values in Table2 and

> **Response(14):** Uncertainty is discussed and quantified in more detail. Specifically, the uncertainty is considered in two aspects, 1) model error interpolated for the year and 2) extrapolation uncertainty which was already calculated as the uncertainty from gap-filling model selection. The two sources are then combined following the law of error propagation (L244).

*3.Specific comments:*

*Experimental set up and management Table 1: please provide more information about soil properties of the mineral soil cover, and underlying peat layers (carbon content, bulk density, C/N and carbon stock) in a higher resolution for the entire aerated soil depth. Please state how many soil samples were taken per depth and where. Please also add the information of the depth location of the schalter.*

> **Response(15):** A table with a higher resolution of soil characteristics is added Appendix B. The methodology is updated (L138)

*Figure 2: please provide information about the location of the chambers of the control site relative to the main ditch.*

**Response(16):** The distance to the main ditch is added (L128). The distance variated between 25 and 40 meters. The location was chosen to exclude a direct effect of the ditch on the water table in the control sites.

*Please provide data of cuttings days, fertilization events and measurement campaigns for CO2, N2O and CH4 as the growth of the grass and thus GPP strongly depend on time of measurement (days after cutting). Information can be added in Figure 7, Appendix B1, C1.*

**Response(17):** We included the harvesting and fertilization events in the figure 7, appendix C1. Fertilization events where added to the figures in appendix D1 and E1. To account for the influence from plant biomass on the CO2 fluxes, linear relationships between grass height and model parameters ($R_{eco,Tref}$, $GPP_{max}$, and α) were developed (L229)

*Please add information about amount and determination of N und C input through slurry application. Please add information about the determination of the yield (dry mass of the grass).*

**Response(18):** This information is added to the methods and results (L164). From every manure application manure samples were taken. Bulk-density was determined, Total nitrogen (TN) and total carbon (TC) was determined in dry slurry material (3 mg) using an elemental CNS analyzer (NA 1500, Carlo Erba; Thermo Fisher Scientific, Franklin, USA) (L160)

*According to Table 1 Site A und B were grazed. How was carbon import through cattle manure determined? How was the carbon export through grazing determined? Was the yield only determined within the chambers or for the entire grassland? How was the grass height determined?*

**Response(19):** The management of the whole field was grazing, however our field site was fenced off to prevent the mentioned problems. The yield was determined inside the chamber frames, to close the carbon budget. Grass height was estimated using a straight scale with a plastic disk with a diameter of 30cm to determine the top of the grass(L187). The management description is updated in the manuscript (L157)

*Gas fluxes Chambers: Was the location of the frames fixed over the two years? Did the vegetation change within the chambers during the experiment?*

**Response(20):** The frames where fixed trough out the two measurement years (L129). The vegetation though out the years remained dominated by *Lolium perenne.* However in spring there were always other species coming up in the frame. However after the first harvest these species disappeared.

*Please add the transparency of the chambers? Was a correction term introduced due to a reduced transparency?*

**Response(21):** We corrected the PAR values outside the chamber since the acrylic glass of the transparent chambers reflected or absorbed at least 8% of the incoming radiation (L184)

*Please add information about the used sign convention, positive fluxes= loss of carbon? Please add information about the used equation for the calculation of GHG balances and assumptions (harvest is assumed to be released as CO2?, loss of dissolved organic carbon?*

**Response(22):** The methodology is expanded upon. The atmospheric sign convention was used. All C fluxes into the ecosystem where defined as negative (uptake from the atmosphere into the ecosystem), and all C fluxes from the ecosystem to the atmosphere are defined as positive. This also holds for non-atmospheric inputs like manure (negative) and outputs like harvests (Positive). Both harvest and manure input are expected to be released as $CO_2$ again (L233). Dissolved organic carbon was not sampled during the experiment.

*L242ff: what is the accuracy of precipitation data derived from satellite images?*

**Response(23):** The accuracy is nine square kilometer. Giving a precipitation value every 3 hours (L154).

*L243: June 2017 seemed to have received more than the average precipitation June is included in the drought period?*

**Response(24):** The average precipitation in June was higher than average, however this is due two days with heavy rain at the end of the month, ending the drought (L279)

*Figure 4: As there were 3 groundwater measurements per site, it is not clear which groundwater table is presented, average of all 3?, what is the SD of the three wells? How is the variability of groundwater level of the control site? Please explain DRN*

**Response(25):** The presented data is data from the logger in the field site (L289), the other groundwater measurements are manual dip wells, recorded each measurement campaign. The data shown in figure 4 is a good depiction of the situation in the control site. Only close to the ditch (Less than 10 meters) there is a higher groundwater table in the summer and lower in the winter.

*L276: I do not understand the sentence "There is variation.." please clarify.*

**Response(26):** There is difference (variation) between the SSI and control site on the different days in regards to temperature and grass height.

*L327-330: What is meant by uncertainty of 3-16 t CO2 ha-1 yr-1. What is represented by 1.6 t CO2 ha-1 yr-1 I for NEE in 2017?L326-332: What is the difference between annual NEE of 47 t CO2 eq. ha-1 yr-1(L327) and emissions of 62 t CO2 eq. ha-1 yr-1 (L313)*

**Response(27)** This part and other parts are rewritten with updated values to specify the values and uncertainties of $R_{eco}$, GPP and NEE.

*L 334-338: Please provide daily CH4 data.*

**Response(28):** Daily data are added to the manuscript in Appendix D

*Table 2 and Table 3:  Please add uncertainty estimates for all components of GHG balance.*

**Response(29):** Modeling and gap-filling uncertainties are updated and added to table 3 for $R_{eco}$, GPP and NEE.

*L. 380: Reco was lower when the differences of groundwater level was >20 cm*

**Response(30):** Correct, this is adjusted in the manuscript (L406).

*L. 420: N2O emissions are not only driven by fertilization events, but also by soil moisture, which should be differ by the treatment.  Thus, the comparison can be biased by missing peak events.*

**Response(31):**   See response(10) Soil moisture is an important driver for the N2O fluxes from these drained peatland systems. We assume that with the method used we missed peaks induced by fertilization and rewetting (L452). However the comparison between the treatments effects on the basis of the different measurement campaigns still provides insight into the effect of SSI on $N_2O$ emissions.

*L428: please use the same sign convention for all cited references.*

**Response(32):**   The references are updated.

*Technical comments: L310: Please state was the 4 t are, SD?,...,*

**Response(33):** this has been clarified in the manuscript. (L347)

*Please indicate A und B in Figure 6*

**Response(33):** A and B are included in figure 6

*Figure 7: please use colors, which can be clearly distinguished*

**Response(34):** Figure 7 is improved to increase the understandability of the figure.

Reviewer (R#2) comments and author responses and changes to ms bg-2020-230

We highly appreciate the very helpful and constructive comments of the anonymous referee, which helped us to further improve the manuscript. We tried to consider all of them.

Reviewer comments are given in italic and with author responses in normal style

**Sub-soil irrigation does not lower greenhouse gas emission from drained peat meadows**

**by Stefan Weideveld et al.**

*Generally, the manuscript will be of interest for readers of Biogeosciences, and the topic of adequate mitigation strategies for drained organic soils is one of high relevance. While the overall result that there is no difference in GHG emissions of this sub-surface irrigation (SSI)system and the control seems to be robust, there are, in my opinion, still four major issues which need to be solved before the manuscript could be considered for publication in BG:*

> **Response (1)** We thank the reviewer for the positive comments and constructive inputs. This will help us improve the manuscript.

• *The authors appear to be surprised that SSI does not result in lowered GHG emissions, but this "surprise" is rather unfounded as the water table is raised only slightly towards a target level of 60 cm below ground, which I would –in line with the IPCC Wetlands Supplement (IPCC, 2014) –still regard as "deeply drained".*

> **Response (2)** We recognize that part of the questions raised are a result of an inadequate framing of the experiment. The introduction is rewritten to improve the framing of current state of the SSI technique. (L71 -L89) In the Netherlands, the aim of the government is to reduce $CO_2$ emission from peat meadow areas by 1 M t by 2030, from which halve is expected to be achieved with the SSI technique (PBL, 2018). To come to this reduction, an area of 50.000 ha with SSI drainage pipes are planned and a $CO_2$ reduction of 50% is expected from this area. However, the current design of the SSI technique aims to increase the lowest water table while maintaining the agricultural function as "business as usual". Also, this technique has never been validated by measured $CO_2$ emission data. Expectations are based on pilots with only soil subsidence measurements. In these pilots a relation between lowest GWT and soil subsidence is found, therefore the elevation of summer GWT is expected to contribute most to the reduction of $CO_2$ emission. The current set-up tested in our experiment aims to explore the effectiveness of SSI on GHG emission for the first time by measurements, also on a large scale on sites representative for the Frisian peat meadows. Therefore, our hypothesis is based on the state of the SSI technique according to policy in practice (L91). And our set-up was made based on the current policy status rather than the scientific exploration of the optimal use of rewetting to mitigate the emissions.

• *According to Figure C1, there were partially only 7 measurement dates for N2O in 2017 and afterwards a gap of five months. Given the highly episodic nature of N2O fluxes, this is absolutely inadequate for the calculation of annual balances in a strongly fertilized grassland.*

**Response (3):** In 2017 we experienced infrastructural constraints to measure $N_2O$ fluxes more frequently. The extended winter gap is a consequence of mal-functioning of the Picarro 2508 under field conditions with low temperature. We agree that 7 flux days and 90 measurements are too few for year budget estimation. We present an average measured N2O flux in table 4 (Page 21). The methods and results will be adjusted so that it becomes clear that measurements of 2017 are a rough estimation based on average fluxes from 7 flux days (L78). However, we believe that the measured data is still valuable for evaluating the $N_2O$ emissions under influence of SSI. The results show no structural higher or lower $N_2O$ emissions between the control and SSI sites. The measured data fits our expectations and references of these types of systems. Clarification be added to methodology and discussion (L494) to stress the low temporal resolution of our measurements, and daily measured data will be presented. The moments between frost and thaw was measured for Farm B and C in the beginning March 2018. However due to technical difficulties with low temperatures and the gas measure equipment these moments were still sparse.

• *For the interpolation of GPP, all measurement campaigns have been pooled for 2017 and harvests have not been accounted for when interpolating GPP despite the large influence of above ground biomass on maximum photosynthetic rates.*

> **Response (4):** We appreciate the reviewer's suggestions for improved gap-filling strategies. The data for GPP gap-filling was available for a recalculation of the GPP balance for 2017 for a better estimate of the GPP. Pooling of measurement campaigns is improved based on conditions during the measurements. The amount of biomass and the harvest are key in understanding the GPP flux. We have included the harvest moments in the interpolations of GPP and corrected for the Interpolated Reco.(L229)

**Title and assumption that this specific SSI system would lower GHG emissions**

*The SSI system studied here has a target water level of -60 cm. Given the limited hydraulic conductivity of the peat and the "exit resistance" of the pipes, a water level of -60 cm in the ditches results in even deeper field water levels in summer. This target seems to be based on the assumption that CO2emissions originate from deeper peat (see below). Thus, the authors state that a WT rise of 6-18 cm in summer compared to an even lower level "unexpectedly" (line 22) or "contrary to our expectations" (line 29) does not lower GHG emissions. In my opinion, this is absolutely no surprise, but should be expected as laboratory studies often show highest respiration rates at medium water content and as field studies, on average, showed an asymptotic rather than a linear response of CO2 emissions to water table depth (too dry, no more peat exposed, Tiemeyer et al., 2020).*

> **Response (5)**: See response(2) for a full response. This is not our own expectation, but the expectations are based on previous pilot studies and now common accepted in policy.

*Thus, the title needs to be changed to "Sub-soil irrigation with target water levels of 60 cm does not lower carbon dioxide emissions from drained peat meadows" or something similar, as the experiments do not allow for conclusions on SSI in general. Further, if the authors are really surprised by their results, they will need to convince the reader why. In this context, it also needs to be discussed why such low target water levels have been chosen at all. At least for meadow use as in 2018, such low water levels are technically not needed when adequate machinery (low weight, double tyres, etc.) is used.*

**Response (6):** A change will are made in the title (L1) and conclusion (L525), to clarify that the current design of SSI is the commonly applied compromise between additional drainage and increased infiltration during summer and that this technique may fall short to have a significant effect on the GHG balance.

**Peat layers below -70 cm contribute most to GHG emissions**

*In the introduction, there is no reasoning why this should be the case at all. Many studies have shown that top soils show higher respiration rates than subsoils e.g. due to higher nutrient contents or generally more favourable conditions for microbial activity(e.g. Bader et al., 2018). This is indeed briefly discussed on page 22, but the whole "story"of the manuscript (and probably also the design of the sub-surface irrigation system) builds on this assumption. Thus, either it needs to be substantiated by peer-reviewed (!) literature, or the manuscript needs to be restructured based on more adequate hypotheses.*

**Response (7):** We agree with the reviewer that the manuscript needed a clear distinction between current knowledge in peatland sciences and (current) assumptions of land authorities and Dutch governmental institutions responsible for emissions reporting from peatlands. In our own scientific reporting (van den Berg et al. 2018) we show that the top 20 cm of peat revealed the highest $CO_2$ production potential. In contrast, the current estimation methods in the Netherlands make no use of $CO_2$ flux data but rely on soil volume – soil carbon models. It is assumed that soil subsidence is quasi 1:1 related to carbon losses in form of $CO_2$ without taking volume changes of the peat and changes in the carbon density into account. Based on that 1:1 soil subsidence-soil carbon relationship it has been inferred that soil subsidence is stronger when groundwater resides during summer (L71 -L89)

**Frequency of N2O flux measurements**

*According to FigureC1, there seem to be only 7 measurement dates for N2O in some cases in 2017, then a gap of more than 5 months in winter and finally a further gap of two months at the end of the study period. This contradicts the text that N2O was measured at each campaign, i.e. supposedly bi-weekly in summer and monthly in winter(page 8). If Figure C1 is actually correct, this data may not be used for the calculation of annual balances as effects of fertilisation cannot be captured adequately with such a low temporal resolution. Further, I would suspect that the first fertilisation event took place before April and was thus missed by the campaigns. In any case, fertilisation dates should be indicated in Figure C1.*

*Even more important, it is well-known that high N2O emissions may occur when temperatures change between frost and thaw(e.g. Koponen and Martikainen, 2004), especially under wetter conditions, and that maximum N2O fluxes of drained peatlands may occur in winter also under temperate climatic conditions (e.g. Flessa et al., 1988). Therefore, the authors should refrain from calculating annual balances from a dataset without winter data. The N2O data could, however, be used to compare treatment effects on the basis of campaigns. In consequence, this means that GHG balances cannot be calculated from the presented data, but only C balances.*

**Response (8):** See response (3) for the elaboration about the $N_2O$ choices that were made in the manuscript and the changes that we made.

***GPP modelling***

*In my opinion, pooling all summer data as done for 2017 is not an adequate gap-filling strategy as GPP max and $\alpha$ strongly depend on vegetation development. This strategy of pooling might be valid for (semi-)natural vegetation, but no for intensively used grasslands with frequent harvests.*

*Further, it seems that parameters are generally interpolated across harvests which does not capture the effects on GPP, which should be very low after harvests. Harvests are unfortunately not indicated in Figure 7 and Appendix B. I would strongly suggest using an interpolation approach suited for highly managed systems (e.g. Eickenscheidt et al., 2015). If this should not be possible due to inadequate PAR ranges during measuring campaigns, only campaign data (instead of annual balances) may be evaluated.*

> **Response (9):** We re-calculated the GPP for the 2017 and 2018 campaign-wise with improved pooling of campaign-wise data and the inclusion of the cutting events where the GPP will be reduced. The GPP estimation has been improved with better parameter fitting. The influence from plant biomass on the $CO_2$ fluxes is now accounted based on linear relationships between grass height and model parameters ($R_{eco,Tref}$, $GPP_{max}$, and $\alpha$) (L229). The harvest dates are included in figure 7 and in appendix C to visualize these moments. And to give a better estimate for the total emission.

***Further comments***

*• Line 59: Better cite the most recent Dutch inventory data instead of an "old" (2009) paper.*

> **Response(10):** The most recent Dutch inventory is used to have an indication of the national emissions from drained peatlands. (L58)

*Table 1:*

*• Details (e.g. SOC, clay content) on the "mineral top layer" would be helpful.• Soil properties averaged for 0 to 70 cm are not really informative, better provide data on the top soil and on depths where the water level/moisture changes actually occurred.*

> **Response(11):** We agree that the current table is inadequate. A table is added to the appendix B to provide additional information on the soil properties for the mineral cover layer, the schalter layer, the degraded peat layer and the less degraded peat layer. The mineral content was determined, however the fractions of the mineral top layer where not determined.

*• How comparable are SSI and control when they partially strongly differ in SOM content (location D) or C:N ratio (location A)?*

> **Response(12):** The differences in organic matter is largely due to the thickness of the mineral top layer. However, for the soil organic carbon stock is of a similar size for both sites. The soil organic matter in table 1 is indicated as g/l soil.

*• Data on hydraulic conductivity or at least on the degree on decomposition are needed to discuss the contrasting hydrologic effects of SSI at the four locations.*

> **Response**(13): The hydraulic conductivity was not measured during the experiment. However, the dip wells that we used to measure the water table for the different frames could give an indication for the processes in the field. Only one location had a good horizontal water flow in the peat layer, that was location B. Location A saw a strong effect of the SSI in the water table. However, in some places there was a large difference in the water table at different distances from the pipes.

*• Does the "schalter" layer have any effect on the sites' hydrology?*

> **Response(14):** Schalter is known to limit vertical water flow, due to its laminated structure. However, there is little documented about the properties and processes (L108). In our case, the locations with "schalter" seem to have lower effects from the SSI

*• Line 140: How was the C-export actually determined? For the frames (line 166 ff) or for the whole field as it is implied here? Were the frames fenced off from grazing?*

> **Response(15):** The experiment sites were fenced off from grazing. The C-Export was determined inside the frames. However, on harvest days the whole experimental site was cut. (L157)

*• If the C-export had to be excluded from statistical analysis, how could the GHG balance containing the C-export be analysed?*

> **Response(16):** With regard to the uncertainties on C-export and CH4/N2O budgets, the statistical analysis is now focused only on annual NEE and measured CH4 and N2O fluxes.(L244 – L260)

*Line 145: Flux measurements and modelling*

*While I understand that not all details can be provided to limit the length of the manuscript, lots of information is missing which would allow assessing the quality of the data.*

*• Were the chambers cooled and vented for pressure equilibration?*

> **Response(17):** The chambers where not cooled. The pressure inside was equilibrated when placing the chamber on the frames. (L178)

*• Was PAR outside the chamber corrected for the light transmittance of the chamber before interpolating GPP? Which light transmittance was assumed/measured?*

> **Response(18):** We corrected the PAR values outside the chamber since the acrylic glass of the transparent chambers reflected or absorbed at least 8% of the incoming radiation. (L184)

*• Was there any quality control procedure for flux calculation (linearity, outliers, leakage...)?*

> **Response(19):** Each flux was checked, for the dark measurements a only fluxes with a R-squared of 0.99 or higher where used. For the light measurements the majority of the fluxes that were used had a R-

squared of 0.95. The exception where the fluxes with slopes close to zero or zero (equilibrium between gross primary production – GPP – and $R_{ECO}$) were not discarded.

• *Was there any minimum temperature difference within one Reco campaign to avoid artefacts due to extrapolation of Eq. 2?*

> **Response(20):** There was no minimum temperature difference set within the Reco campaign. The measurement campaigns where planned to have a range in light variation for the GPP calculations, this resulted in a good temperature range during the day. (L208)

• *Which Reco(nearest?) was subtracted from NEE to yield GPP (line 192)?*

> **Response(21):** The Reco closest in time was used for subtraction of NEE to yield GPP. During the campaigns light and dark measurements were always conducted in the same time frame. (L212)

• *The unit of $\alpha$ is wrong (line 200), it should be mg CO2-C m-2h-1/ μmol m-2s-1*

> **Response(22):** Is adjusted in the material and methods. (L220)

• *Why did you choose to interpolate parameters and not weighted fluxes(line 204)?*

> **Response(**23): We have updated the interpolation method and adjusted in the manuscript, for the recalculations we used weighted fluxes instead of interpolating the parameters since it has been suggested to be better (Hoffmann et al. 2015). Extrapolated values at times between two adjacent models are weighted averages of the estimates from these two models, where the weights are temporal distances of the extrapolated time spots to both of the measurements (L227)

• *Why did you use GPPmax and not GPPopt(Falge et al., 2001)which is less susceptible to extrapolation errors?*

> **Response (24):** According to Falge et al. (2001), GPPmax (or saturation value of GPP) has less explanatory worth for real systems since PAR will not reach infinite, therefore the author switched to GPPopt which provides a reference value at a certain PAR level. GPPmax was used in numerous fluxes modeling works, and we did not find argument from literature stating significantly larger uncertainty from the use of GPPmax. Thereby, if a GPPopt is used, there should have enough data in a specific PAR value (e.g. 2000 μmol m$^{-2}$ s$^{-1}$). With eddy covariance (where this gap filling paper of Falge et al. is written for) this is not a problem, but with chamber measurement data is limited. The accuracy will therefore be better to fit the light response curve with the GPPmax.

**Line 254 ff: "Drainage" and "irrigation" periods**

• *From my understanding, "drainage" and "irrigation" periods are not defined correctly. While it remains unclear which WT (0.5, 1.5 or 3.0 m from the pipe) was used for this calculation, it is of course useful to differentiate whether SSI was dryer or wetter than the control when comparing Reco. However, "drainage" and "irrigation" periods can only be identified by using absolute heads, i.e. by comparing the field WT to the ditch water level!*

**Response(25):** The definition of these periods is clarified (L292). 'Drainage' periods refer to moments when there is drainage to the ditch and 'irrigation' periods refer to moments with water infiltration from the ditch to the field. Here the aim is to differentiate between SSI and control.

*• In this context, it also remains unclear why the SSI system works better at some of the fields in terms of hydrology –is there always enough water in the ditches, how is the hydraulic conductivity or at least the degree of decomposition of the peat, or are there strong differences in WT at 0.5, 1.5 and 3.0 m difference to the pipes which could be used to deduce information on the hydraulic conductivity?*

**Response(26):** The water table in the ditch was maintained at a level between -60 to -20 cm from the soil surface. It was never a limiting factor for the functioning of SSI. The hydraulic conductivity of the peat soil was not measured during the experiment. However the functioning of the SSI gives an indication of the conductivity. This is closely related to the type of peat present. Farm A, C and D all have Sphagnum peat, with the layer where the pipe is present being moderately decomposed (H5-H7). We suspect there are some macro cracks in the peat soil of farm A, that help infiltration. For location B the peat soil consists of Alder peat. The layer where the pipe is present is moderately decomposed but with a large presence of wood/branches. For this location the SSI seems to work best. With a strong drainage and infiltration effect.

*• Furthermore, it is rather difficult to compare results to other studies. Therefore, please give numbers for the mean and the summer mean water level.*

**Response(27):** The mean annual average GWT table is added in Table 2 to increase the comparability between the different sites and to other studies.

***Table 2 and Table 3***

*•Should be merged and N-fertilisation should be added. •Uncertainties should be added.*

**Response(28):** The tables where not merged, the readability was in proved, to make a division between the Net ecosystem carbon balance (NECB) and the CH4 and $N_2O$ emissions Modeling and gap-filling uncertainties have been added to $R_{eco}$, GPP and NEE. (Table 3).     N-Fertilization is added to the methodology (L164)

*• **Line 425 ff:** There are some comparisons to other studies, but the authors do not try to explain the differences in emissions between their sites.*

**Response(29):** The differences between the sites are largely because of the soil conditions (Appendix B). The locations with a mineral topsoil seem to respond stronger to drought. Furthermore, there was a difference between the starting conditions of the sites. The sites A and B where grazed before the experiment and site C and D where only mown Table 1. This resulted in a difference in the grass structure, where the grazed grass forms a more dense vegetation structure than the mown grass.

*• **Line 439 :** How do you know that moisture conditions were optimal in 2017?*

**Response(30):** The indication of 'optimal' come from observation of the conditions in the field, for example the grass growth that we observed during the field experiment. This was also determined in contact with the farmers who judged a better year for grass growth. We will reconsider the wording 'optimal', but the point was that we expect that the moisture levels were not a limiting factor during this summer period. (L489)

• *Line 451: What do you mean with "abnormal data points"?*

**Response(31):** The "abnormal data points" refer to measurement that did not fit into the temperature dependent function of $R_{eco}$ or light response curve of GPP, due to the extreme drought that limited soil respiration's response to higher temperature, or reduced the photosynthetic rate. Wording has been adjusted to avoid confusion (L501).

• *Line 486 : Effects of land-use intensity and land-use history should be discussed in the context of general emission level (section 4.3) as these aspects do not fit to the section "costs and benefits SSI".*

**Response (32):** We chose to discuss about land use in the 'cost and benefit' section because of the possibility of SSI to be beneficial for the intensive land use. Due to the increased load boarding capacity of the fields and the drainage in Spring and Autumn, it is possible to extend the periods that the field can be managed. We consider this as a possible benefit from the SSI, however we didn't observe this during the experiment.

• *Besides methodological issues, the manuscript seems to be hastily prepared which results in many inaccuracies especially regarding the references (list might not be exhaustive): • Several references mentioned in the text are not in the list of references (Hoffmann et al., 2015, Tiemeyer et al., 2020)• One reference appears twice (Berglund and Berglund, 2011)•References are incomplete (Couwenberg, 2009, Tanneberger et al., 2017)*

**Response(33):** The references have been updated.

• *Generally, there is some tendency to cite non-peer reviewed literature (Joosten and Clarke, 2002, Joosten, 2009, Jurasinski et al., 2016, Hendriks et al., 2007b, Hoving et al., 2015, van den Akker et al., 2008, van den Born et al., 2016). In many cases, peer-reviewed papers could easily be found and should be cited instead.*

**Response(34) :** The current references will be updated. The choice for the non-peer reviewed literature is largely due to the current condition that many of the decisions made for the SSI-experiment by the local government were based on these references. And some indicate the aim of the national and provincial government to implement SSI on a large scale as a way of mitigating problems that occur with management of these Peat meadows.

• *Furthermore, table and figure headings are often very brief or contain abbreviations, sometimes also such which are not used in the manuscript(e.g. location "Ger" in Appendix B).*

**Response(35):** The table and figure headings are expanded upon the improve the understandability of the figures and the abbreviations will be written full out in the headings.

[revised manuscript text omitted]

---

## Author Response (AR2)

**Author's response**

Reviewer (R#1) comments and author responses and changes to ms bg-2020-230

We highly appreciate the very helpful and constructive comments of the anonymous referee, which helped us to further improve the manuscript. We tried to consider all of them.

Reviewer comments are given in italic and with author responses in normal style

*The authors improved their manuscript. They now describe their reason for this study adequately. However, I have still several concerns that should be addressed before reconsideration for publication in Biogeosciences. Especially the discussion part needs to be revised as some discussion points are not well-thought-out and references are imprecise or wrongly cited.*

*N2O data*
*Hypothesis 3 cannot be answered with this data set. With a low frequency of sampling, it cannot be excluded that peak emission were missed. In order to answer this question I would suggest to have at least weekly N2O measurement (with higher resolution in situation with high N2O emission risk or continuous data from Eddy-Covariance systems. As the authors stated in their response, it was intended to measure N2O at a higher frequency. Unfortunately, it failed. I would suggest deleting hypothesis 3 instead of trying it to answer it with in inadequate data set. I suggest considering to change ghg to carbon emissions in the title. In the abstract the frequency for N2O measurements are missing, but the authors stated, that the N2O emission were lower in 2017 than in 2018. In addition, N2O data are included in the ghg budget (241, 455). This implies that there is a sufficient data base for calculation of N2O budget. In addition, N2O emissions could also be underestimated by missing peak emission in summer (466). Therefore, it is not easy to estimate the contribution of N2O to ghg budget. Please add number of N2O (and CH4) measurements in Table 4.*
*According to the method section site D 17 measurement campaigns took place. However, I count 13 data points for SII.*
*L462:The resolution of N2O data is too low to detect peak emissions. For N2O peaks not only fertilization events but also soil humidity is essential. The soil humidity is changed by treatments and could differ. The soil moisture in one treatment could be high enough for N2O emissions, but not in the control. The N2O emissions could also occur one different days due to different moisture regimes in soil. Therefore, it is not possible to state that no peak emission were missed.*

> **Response:** We recognize that the data for N2O is insufficient to make a good estimate for a N2O balance. We changed our title to Carbon emissions, and removed hypothesis 3 from the manuscript. Now the focused for N2O is on the direct comparison of measured fluxes between SSI and control, which is sufficient in showing the absent of treatment effects. It is also addressed in discussion that we have too few measurements to engage gap-filling and statistical analysis of year budgets for N2O.

*Experimental set up.*
*My second concern is the frequency of grass mowing within the chambers in 2017. The grass was cut 8 times, which is considerably higher than in 2018 and for the whole pasture (4-5 times) and as generally applied for intensively used pastures. For me it is not clear why the grass was cut in 2017 at this high frequency. The frequency of mowing affects the grass development and thus yield. Therefore, comparison of yield may be biased by different management and not only by climatic conditions, as the authors interpret it.*

> **Response:** We made an additional cut in the start of May 2017, because of the fast grass growth and grass height exceeding 30 cm. which resulted in an out of sync mowing regime compared to the surrounding farmland, we adjusted for this. The month of September and

October were extremely wet, this resulted that most farmers decided not to harvest the grass. We however included this harvest. At the end of the year, there was a relatively high amount of biomass remained in the field, so we added an extra harvest to close the Carbon balance of the year. This was not necessary during the drought of 2018.

*I still wonder why a considerably CO2 uptake can be observed after grass cuts, as the CO2 uptake is generally considerably reduced after mowing not only for organic soils, but also for mineral soils (Beetz et al. 2013, Poyada et al. 2016, Eickenscheidt et al. 2015, Schmitt et al., 2010). This question was also addressed in the first review. One reason may be that the modelling/gap-filing is of too low quality to capture the management events of the grassland or the grass was only cut slightly, so considerably amount of CO2 can still be taken up after harvest. However, other grassland with 4-5 cuts per year show the reduction of CO2 uptake after mowing.*

**Response:** This is now addressed in the Method section with **"**Models developed for the campaign before harvesting were then corrected using the slopes of the linear regressions as the models after the harvest to be applied in the extrapolation. The loss of biomass was therefore accounted according to lowered grass height, different from the studies where model parameters are to zero after harvest (e.g. Beetz et al. 2013). **"**

We chose not to set the parameters (GPPmax, alpha, etc.) to near zero after harvest as did in the cited studies, because at the start of the experiment we had some moments that we measured after harvesting. There was indeed a drop in $CO_2$ uptake, however not to 0. And the biomass was not completely removed but with a small amount of residual with grass height of $5 - 7$ cm. The grass height has good agreement with the GPP estimations; therefore, we used this relationship to correct for the harvest rather than a manual reset.

*Discussion*

*Especially, the whole discussion session needs to be carefully revised. The argumentation and wording are not always straightforward or even false. It is not clear, what the land use, peatland type, management and origin of the cited sites are. In addition, it is often not distinguished between field experiment or laboratory studies.*

**Response:** We worked through the discussion to rewrite the argumentation and the wording and to make sure that the cited studies are properly placed and described. As we recognize that there was a lack of consistency between our arguments with the supporting references in the previous version of the manuscript.

*Here are some examples:*
*Line 127ff:*
*I guess that the site had been drained for long-term (highly decomposed material line 444). Thus, the top 30-40 cm have under oxidized condition for long-term and easily decomposable organic material may have been already decomposed. Accordingly, relative stable organic matter may have been accumulated at the top 30-40 cm which is in contrast to the argumentation in line 127ff.*

**Response:** As a pilot tryout for this study, we incubated soils from the field from different depths under ideal oxidation conditions, we did not observe lowered decomposition activity from topsoil layers or increased emissions from deeper/more pristine layers. Furthermore, our fluxes measurements did not show a low emission in situation where only the topsoil was exposed and drained. Therefore, our arguments did not go to the direction of the reviewer's assumption.

*In addition, it is not clear if the water content of deeper layers is saturated and how the O2 saturation is, as these parameters are site-specific. The annual averaged soil temperature at deeper layers may be the same as at top layers.*

**Response:** We only measured the moisture content for the deeper layers with physical samples. Saturation of the deeper layers and the O2 saturation would be the factors that are of interest to follow and see how they are influenced by SSI. However, we did not include this in our setup.

*Line117ff:*
*After Tiemeyer et al. 2020 there is a general dependency of groundwater table and CO2-emissions rates for all land use classes, most strongly at water levels between -20 and -50. The dependency of deeper groundwater level on CO2 emissions is less clear. The findings of the authors are in agreement with recent literature.*

**Response**: We removed the statement that our findings are contrary to the general assumption

*L 405-409: here are results presented not discussion*

**Response:** This is removed from the discussion.

*L422-423:*
*The soil moisture data shows differences between 33 and 90% (Appendix). Not sure, if I would classify these differences as small variations.*

**Response**: "This lack of effect is explained by the fact that there is only a small difference in soil moisture values above the GWT" Is an explanation of what was found in literature, by Lafleur et al., (2005);Nieveen et al., (2005);Parmentier et al., (2009).

*L423-425:*
*I can only guess the meaning of this sentence. Please reformulate.*

**Response:** We reformulated the sentence, what we were trying to say is that ,the lower $CO_2$ emissions reported with structurally elevated GWT often have a vegetation and land use that are more adapted for the higher water table.

*L 427: Please reformulate the sentence. What is effect size*

**Response**: We reformulated the sentence, "treatment effect on measured $R_{eco}$"

*L 473:*
*In Tiemeyer et al. 2016 it can be clearly seen that there are two sites with NEE > 60 t CO2 (Fig 1), so there have been higher values measured before. The cited values (Tiemeyer at al 2020) are the new German EF based on published data. The German EF are self-evident lower than the highest measured values.*

**Response:** Agreed, we adjusted this in the text.

*Soil moisture*

*The authors now include soil moisture from one sampling day in August 2017. Unfortunately, the groundwater was quite similar between treatments in 2017. I asked for the soil moisture data from sensors. If these data are not included in the manuscript, than they can be deleted from the material and method section.*

> **Response**: The soil moisture sensors where often the cause of the malfunction of the sensor station. In the summer the sensor would give faulty data, because of the depth of the sensor and the soil properties of the clay layer. This resulted in incomplete time series, that is why we decided not to include this in manuscript. It is now deleted form the material and method section.

*T soil is used for the modeling. How is the data coverage of the 5 cm soil data, as it can be expected that malfunction of sensors was both for soil moisture and soil temperature? When was the Hobo sensor installed? Which data was used for modelling 5 cm or 10 cm Hobo data? How much uncertainty add the missing soil data to the NEE? Which data was used, when there was a data gap?*

> **Response:** The sensors were installed at the start of the experiment, and the problems occurred for both sensors quite early into the experiment. That is why we installed multiple sensors that were more robust to the field sites. The main data used for the modeling of NEE is from the 5 cm sensor of the main station, but if there was a malfunction data from the additional sensors that were installed was used.

*Soil properties are now included in the Appendix. I would suggest to incorporate the information about C content (g/kg) of different layers in Table 1. The unit g/l is not often used. It is rather carbon stock than carbon content?*

> **Response:** Table 1 and appendix are now combined to provide additional information. In the table we display carbon content and carbon stock. This will help to understand the conditions in the field and the properties of the high organic clay cover that is present in our field sites.

*Please provide also errors of NECB and c-export and display them in Table 3 (Also for Manure)*

> **Response:** The errors are now included in the table for the NECB, c-export and manure.

*Schmitt, M., M. Bahn, G. Wohlfahrt, U. Tappeiner & A. Cernusca. 2010: Land use affects the net ecosystem CO2 exchange and its components in mountain grasslands, Biogeosciences, 7, 2297-2309*